# THE GAMBLER'S PROBLEM AND BEYOND

**Baoxiang Wang**
Department of Computer Science and Engineering
The Chinese University of Hong Kong
bxwang@cse.cuhk.edu.hk

**Shuai Li**
John Hopcroft Center for Computer Science
Shanghai Jiao Tong University
shuaili8@sjtu.edu.cn

**Jiajin Li**
Department of SEEM
The Chinese University of Hong Kong
jjli@se.cuhk.edu.hk

**Siu On Chan**
Department of Computer Science and Engineering
The Chinese University of Hong Kong
siuon@cse.cuhk.edu.hk

## ABSTRACT

We analyze the Gambler's problem, a simple reinforcement learning problem where the gambler has the chance to double or lose their bets until the target is reached. This is an early example introduced in the reinforcement learning textbook by Sutton & Barto (2018), where they mention an interesting pattern of the optimal value function with high-frequency components and repeating non-smooth points. It is however without further investigation. We provide the exact formula for the optimal value function for both the discrete and the continuous cases. Though simple as it might seem, the value function is pathological: fractal, self-similar, derivative taking either zero or infinity, not smooth on any interval, and not written as elementary functions. It is in fact one of the generalized Cantor functions, where it holds a complexity that has been uncharted thus far. Our analyses could lead insights into improving value function approximation, gradient-based algorithms, and Q-learning, in real applications and implementations.

## 1 INTRODUCTION

We analytically investigate a deceptively simple problem, the Gambler's problem, introduced in the reinforcement learning textbook by Sutton & Barto (2018), on Example 4.3, Chapter 4, page 84. The problem setting is natural and simple enough that little discussion was given in the book apart from an algorithmic solution by value iteration. A close inspection will however show that the problem, as a representative of the entire family of Markov decision processes (MDP), involves a level of complexity and curiosity uncharted in years of reinforcement learning research.

The problem discusses a gambler's casino game, where they places multiple rounds of betting. The gambler gains the bet amount if they win a round or loses the bet if they lose the round. The probability of losing each round is $p \geq 0.5$, independently. The game ends when the gambler's capital reaches either their goal of $N$ or $0$. In each round, the gambler must decide what portion of the capital to stake. In the discrete setting this bet amount must be an integer, but it can be a real number in the continuous setting. To formulate it as an MDP, we denote state $s$ be the current capital and action $a$ the bet amount. The reward is $+1$ when the state reaches $s = N$, and zero otherwise.

Our goal is to solve the optimal value function of the problem. We first give the solution to the discrete Gambler's problem. Denote $N$ as the target capital, $n$ as the current capital (which is the state in the discrete setting), $p > 0.5$ as the probability of losing a bet, and $\gamma$ as the discount factor. The special case of $N = 100, \gamma = 1$ corresponds to the original setting in Sutton and Barto's book.

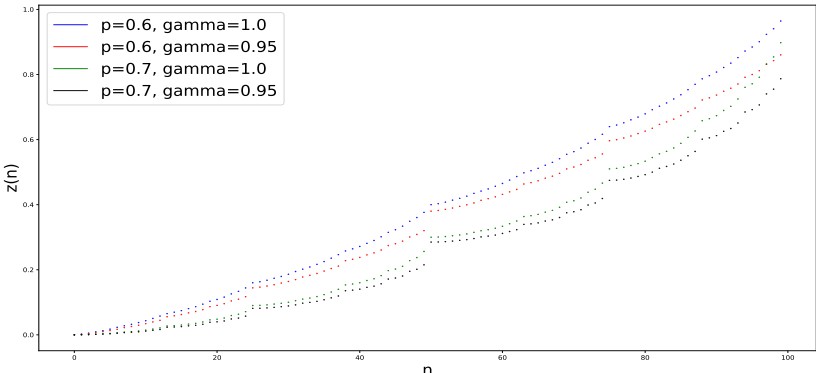

Figure 1: The optimal state-value function of the discrete Gambler's problem.

**Proposition 1.** *Let $0 \leq \gamma \leq 1$ and $p > 0.5$. The optimal value function $z(n)$ is $v(n/N)$ in the discrete setting of the Gambler's problem, where $v(\cdot)$ is the optimal value function under the continuous case defined in Theorem 12.*

The above statement amounts the discrete problem to the continuous problem by a uniform discretization. The rest of the discussion will be on the more general continuous setting. In the setting, the target capital is 1, the state space is $[0, 1]$, and the action space is $0 < a \leq \min\{s, 1 - s\}$ at state $s$, meaning that the bet can be any fraction of the current capital as long as the capital after winning does not exceed 1. We state the optimal value function below and intuitive description of the value function later in this section.

**Theorem 12.** *Let $0 \leq \gamma \leq 1$ and $p > 0.5$. Under the continuous setting of the Gambler's problem, the optimal state-value function is $v(1) = 1$, and*

$$v(s) = \sum_{i=1}^{\infty}(1 - p)\gamma^i b_i \prod_{j=1}^{i-1}((1 - p) + (2p - 1)b_j) \tag{1}$$

*for $0 \leq s < 1$, where $s = 0.b_1 b_2 \ldots b_\ell \ldots_{(2)}$ is the binary representation of the state $s$.*

Next, we solve the Bellman equation of the continuous Gambler's problem. In the strictly discounted setting $0 \leq \gamma < 1$, the solution of the Bellman equation $f(0) = 0$, $f(1) = 1$,

$$f(s) = \max_{0 < a \leq \min\{s, 1-s\}} (1 - p)\gamma f(s + a) + p\gamma f(s - a)$$

is $f(s) = v(s)$ the optimal value function (Proposition 21).

This uniqueness does not hold in general. If the rewards are not discounted, the solution of the Bellman equation is either the optimal value function, or a constant function larger than 1.

**Theorem 22.** *Let $\gamma = 1$, $p > 0.5$, and $f(\cdot)$ be a real function on $[0, 1]$. $f(s)$ solves the Bellman equation if and only if either*

- *$f(s)$ is $v(s)$ defined in Theorem 12, or*

- *$f(0) = 0$, $f(1) = 1$, and $f(s) = C$ for all $0 < s < 1$, for some constant $C \geq 1$.*

Under the corner case of $\gamma = 1$, $p = 0.5$ (where the gambler do not lose capital in bets in expectation), the problem involves the midpoint concavity (Sierpiński, 1920a;b) and Cauchy's functional equation. The measurable function that solves the Bellman equation is uniquely $f(s) = C's + B'$ on $s \in (0, 1)$, for some constants $C' + B' \geq 1$. Additionally, under Axiom of Choice, $f(s)$ can also be some non-constructive, non Lebesgue measurable function described by a Hamel basis (Theorem 27 and its lemmas).

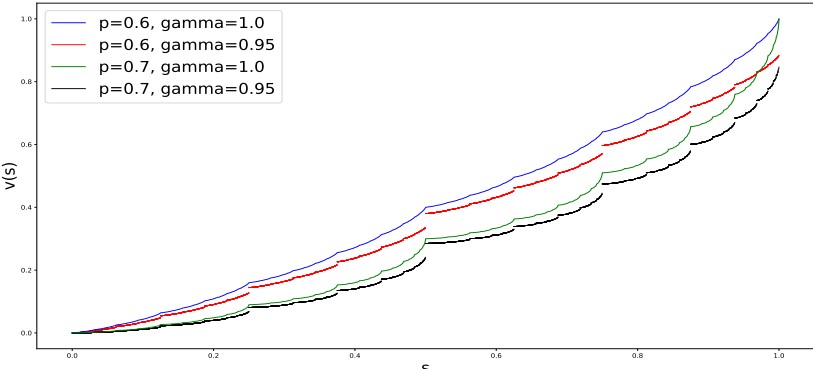

Figure 2: The optimal state-value function of the continuous Gambler's problem.

Though the description of the Gambler's problem seems natural and simple, Theorem 12 shows that its simpleness is deceptive. The optimal value function is fractal and self-similar, and non-rectifiable (see Corollary 14 and Lemma 8). It is thus not smooth on any interval, which can be unexpected when a significant line of reinforcement learning studies is based on function approximation like discretization and neural networks. The value function (1) can neither be simplified into a formula of elementary functions, which introduces difficulties in understanding it. The function is monotonically increasing with $v(0) = 0$ and $v(1) = 1$, but its derivative is 0 almost everywhere, which is counterintuitive. This is known as singularity, a famous pathological property of functions. $v(s)$ is continuous almost everywhere but not absolutely continuous. Also when $\gamma$ is strictly smaller than 1, it is discontinuous on a dense and compact set of infinitely many points. These properties indicate that assumptions like smoothness, continuity, and approximability are not satisfied in this problem. In general, it is reasonable to doubt if these assumptions can be imposed in reinforcement learning. To better understand the pathology of $v(s)$, we analogize it to the Cantor function, which is well known in analysis as a counterexample of many seemingly true statements (Dovgoshey et al., 2006). In fact, $v(s)$ is a generalized Cantor function, where the above descriptions are true for both $v(s)$ and the Cantor function.

**Intuitive description of $v(s)$.** All the statements above require the definition of $v(s)$. In fact, in this paper, $v(s)$ is important enough such that its definition will not change with the context. The function cannot be written as a combination of elementary functions. Nevertheless, we give an intuitive way to understand the function for the original, undiscounted problem. The function can be regarded as generated by the following iterative process. First we fix $v(0) = 0$ and $v(1) = 1$, and compute

$$v(\frac{1}{2}) = pv(0) + (1 - p)v(1) = (1 - p).$$

Here, $v(\frac{1}{2})$ is the weighted average of the two "neighbors" $v(0)$ and $v(1)$ that have been already evaluated. Further, the same operation applies to $v(\frac{1}{4})$ and $v(\frac{3}{4})$, where $v(\frac{1}{4}) = pv(0) + (1 - p)v(\frac{1}{2}) = (1 - p)^2$ and $v(\frac{3}{4}) = pv(\frac{1}{2}) + (1 - p)v(1) = (1 - p) + p(1 - p)$, and so forth to $v(\frac{1}{8})$, $v(\frac{3}{8})$, etc. This process evaluates $v(s)$ on the dense and compact set $\bigcup_{\ell \geq 1} G_\ell$ of the dyadic rationals, where $G_\ell = \{k2^{-\ell} \mid k \in \{1, \ldots, 2^\ell - 1\}\}$. With the fact that $v(s)$ is monotonically strictly increasing, this dyadic rationals determines the function $v(s)$ uniquely.

This iterative process can also be explained from the analytical formula of $v(s)$. Starting with the first bit, a bit of 0 will not change the value, while a bit of 1 will add $(1-p)\prod_{j=1}^{i-1}((1-p)+(2p-1)b_j)$ to the value. This term can also be written as $(1 - p)((1 - p)^{\#\mathbf{0} \text{ bits}} \cdot p^{\#\mathbf{1} \text{ bits}})$, where the number of bits is counted over all previous bits. The value $(1 - p)^{\#\mathbf{0} \text{ bits}} \cdot p^{\#\mathbf{1} \text{ bits}}$ decides the gap between two neighbor existing points in the above process, when we insert a new point in the middle. This insertion corresponds to the iteration on $G_\ell$ over $\ell$.

We provide high resolution plots of $z(n)$, $N = 100$ and $v(s)$ in Figure 1 and Figure 2, respectively. The non-smoothness and the self-similar fractal patterns can be clearly observed from the figures. Also, $v(s)$ is continuous when $\gamma = 1$ while $v(s)$ is not continuous on infinitely many points when $\gamma < 1$. In fact, when $\gamma < 1$, the function is discontinuous on the dyadic rationals $\bigcup_{\ell \geq 1} G_\ell$ while continuous on its complement, as we will rigorously show later.

**Self-similarity.** The function $v(s)$ on $[\bar{s}, \bar{s} + 2^{-\ell}]$ for any $\bar{s} = 0.b_1 b_2 \ldots b_{\ell(2)}$, $\ell \geq 1$ is self-similar to the function itself on $[0, 1]$. Let $s = 0.b_1 b_2 \ldots b_\ell \ldots_{(2)} \in [\bar{s}, \bar{s} + 2^{-\ell}]$, this can be observed by

$$
\begin{aligned}
v(s) &= \sum_{i=1}^{\infty} (1-p)\gamma^i b_i \prod_{j=1}^{i-1} ((1-p) + (2p-1)b_j) \\
&= \sum_{i=1}^{\ell} (1-p)\gamma^i b_i \prod_{j=1}^{i-1} ((1-p) + (2p-1)b_j) + \gamma^\ell \left( \prod_{j=1}^{\ell} ((1-p) + (2p-1)b_j) \right) \\
&\quad \times \sum_{i=1}^{\infty} (1-p)\gamma^i b_{\ell+i} \prod_{j=1}^{i-1} ((1-p) + (2p-1)b_{\ell+j}) \\
&= v(\bar{s}) + \gamma^\ell \prod_{j=1}^{\ell} ((1-p) + (2p-1)b_j) \cdot v(2^\ell(s - \bar{s})).
\end{aligned}
\tag{2}
$$

The self-similarity can be compared with the Cantor function (Dovgoshey et al., 2006; Mandelbrot, 1985), which uses the ternary of $s$ instead in the formula. The Cantor function is self-similar to itself on $[\bar{s}, \bar{s} + 3^{-\ell}]$, when $\bar{s} = 0.b_1 b_2 \ldots b_{\ell(3)}$ and $b_\ell \neq 1$. Both $v(s)$ and the Cantor function can be uniquely described by their self-similarity, the monotonicity, and the boundary conditions.

**Optimal policies.** It is immediate by Theorem 12 and Lemma 8 that

$$\pi(s) = \min\{s, 1 - s\}$$

is one of the (Blackwell) optimal policies. Here, Blackwell optimality is defined as the uniform optimality under any $0 \leq \gamma \leq 1$. This policy agrees with the intuition that under a game that is in favor of the casino ($p > 0.5$), the gambler desires to bet the maximum to finish the game in as little cumulative bet as possible. In fact, the probability of reaching the target is the expected amount of capital by the end of the game, which is negative linear to the cumulative bet.

The optimality is not unique though, for example, $\pi(\frac{15}{32}) = \frac{1}{32}$ is also optimal (for any $\gamma$). Under $\gamma = 1$ the original, undiscounted setting, small amount of bets can also be optimal. Namely, when $s$ can be written in finite many bits $s = b_1 b_2 \ldots b_{\ell(2)}$ in binary (assume $b_\ell = 1$), $\pi(s) = 2^{-l}$ is also an optimal policy. This policy is by repeatedly rounding the capital to carryover the bits, keeping the game to be within at most $\ell$ rounds of bets.

## 1.1 PRELIMINARIES

We use the canonical formulation of the discrete-time Markov decision process (MDP), denoted as the tuple $(\mathcal{S}, \mathcal{A}, \mathcal{T}, r, \rho_0, \gamma)$. That includes $\mathcal{S}$ the state space, $\mathcal{A}$ the action space, $\mathcal{T} : \mathcal{S} \times \mathcal{A} \times \mathcal{S} \to \mathbb{R}^+$ the transition probability function, $r : \mathcal{S} \to \mathbb{R}$ the reward function, $\rho_0 : \mathcal{S} \to \mathbb{R}^+$ the initial state distribution, and $\gamma \in [0, 1]$ the unnormalized discount factor. A deterministic policy $\pi : \mathcal{S} \to \mathcal{A}$ is a map from the state space to the action space. In this problem, $\mathcal{T}(s, a, s - a)$ and $\mathcal{T}(s, a, s + a)$ are $p$ and $1 - p$ respectively for $s \in \mathcal{S}$, $a \in \mathcal{A}$, and $\mathcal{T}$ is otherwise 0.

Our goal is to solve the optimal value function of the Gambler's problem. In this problem, the state-value function is the probability of the gambler eventually reaching the target capital from a state. The definition of the state-value function of an MDP with respect to state $s$ and policy $\pi$ is

$$
f^\pi(s) = \mathbb{E}\left[ \sum_t \gamma^t r_t \Big| s_0 = s, a_t = \pi(s_t), s_{t+1} \sim \mathcal{T}(s_t, a_t), r_t \sim r(s_t), t = 0, 1, \ldots \right].
$$

When $\pi^*$ is one of the optimal policies, $f^{\pi^*}(s)$ is the optimal state-value function. Despite that there may exist more than one optimal policies, this optimal state-value function is unique (Sutton & Barto, 2018; Szepesvári, 2010).

## 1.2 IMPLICATIONS

Our results indicate hardness on reinforcement learning (Papadimitriou & Tsitsiklis, 1987; Littman et al., 1995; Thelen & Smith, 1998) and revisions of existing reinforcement learning algorithms. It is worth noting that similar patterns of fractal and self-similarity have been observed empirically, for example in Chockalingam (2019) for the Mountain Car problem. With these characterizations being observed in simple problems like Mountain Car and the Gambler's problem, our results are expected to be generalized to a variety of reinforcement learning settings.

The first implication is naturally on function value function approximation, which is a developed topic in reinforcement learning (Lee et al., 2008; Lusena et al., 2001). By the fractal property of the optimal value function, that representation of such function must be inexact (Tikhonov, 2014). When discretization is used for value function representation, the approximation error is at least $\mathcal{O}(1/N)$, where $N$ is the number of bins.

**Proposition 19.** *When $N \in \mathbb{N}^+$, $N \geq 4$ is a power of $2$, let $\bar{v}_1(s)$ be piecewise constant on any of the intervals $s \in (k/N, (k+1)/N)$, $k = 0, \ldots, N-1$, then*

$$\int_s |v(s) - \bar{v}_1(s)| \, ds \geq \frac{1}{N} \frac{(2-\gamma)(1-p)\gamma}{1-p\gamma} + o(\frac{1}{N}).$$

Alternatively, when a subclass of $L$-Lipschitz continuous is used to represent $v(s)$, this error is then at least $(1/L) \cdot (1-p)^2 \gamma^2 (1-\gamma)^2 / (4-4p\gamma)$ by the discontinuity of $v(s)$ (Proposition 20). It is worth remarking that despite this specific lower bound diminishes when $\gamma$ is 1, the approximation error is nonzero for an arbitrarily large $L$ under $\gamma = 1$, as the derivative of $v(s)$ can be infinite (Fact 16).

Notably, neural networks are within this family of functions, where the Lipschitz constant $L$ is determined by the network architecture. By the proposition, it is not possible to obtain the optimal value function when a neural network is used, albeit the universal approximation theorem (Csáji, 2001; Dovgoshey et al., 2006; Levesley et al., 2007).

The second implication is by Theorem 12 and Fact 16 that the derivative of $v(s)$ is

$$\lim_{\Delta s \to 0^+} \frac{v(s + \Delta s)}{\Delta s} = 0, \quad \lim_{\Delta s \to 0^-} \frac{v(s + \Delta s)}{\Delta s} = \begin{cases} +\infty, & \text{if } s = 0 \text{ or } s \in \bigcup_{\ell \geq 1} G_\ell, \\ 0, & \text{otherwise.} \end{cases}$$

This imposes that the value function's derivative must not be exactly obtained, as it is 0 almost everywhere, except on the dyadic rationals $G_\ell$, where it has a left derivative of infinity and a right derivative of 0. Algorithms relying on $\partial v(s)/\partial s$ and $\partial Q(s,a)/\partial a$ (Lillicrap et al., 2015; Gu et al., 2017; Heess et al., 2015; Fairbank & Alonso, 2012; Fairbank, 2008; Pan et al., 2019; Lim et al., 2018), where $Q(s,a)$ is the action-value function (Sutton & Barto, 2018), can suffer from the estimation error or even have unpredictable behavior.

In practice, the boolean implementation of float numbers can further increase this error, as all points $s$ implemented are in $G_\ell$ for some $\ell$. A precise evaluation requires all these derivatives to be infinity when a Leabague derivative is used (the average of left and right derivatives), which cannot be obtained a computer system.

The third implication is on Q-learning (Mnih et al., 2015; Watkins & Dayan, 1992; Baird, 1995), by Theorem 22 and its supporting lemmas. It is proved that when $\gamma = 1$, Q-learning has multiple converging points, as the Bellman equation has multiple solutions, namely $v(s)$ and

$$f(0) = 0, f(1) = 1, \text{ and } f(s) = C \text{ for all } 0 < s < 1,$$

for some constant $C \geq 1$. Therefore, even when the Q-learning algorithm converges, it may not converge to the optimal value function $v(s)$. In fact, as the solution can be either the ground truth of the optimal value function, or a large constant function, it is easier to approximate a constant function than the optimal value function, resulting in a relatively lower Bellman error when converging to the large constant.

This challenges Q-learning under $\gamma = 1$ when the return (cumulative reward) is unbiased. Though the artificial $\gamma$ is originally introduced to prevent the return from diverging, it can be also necessary to prevent the algorithm from converging to a large constant in Q-learning, which is not desired.

## 2    DISCRETE CASE

The analysis of the discrete case of the Gambler's problem will give an exact solution. It will also explain the reason the plot on the book has a strange pattern of repeating spurious points.

The discrete case can be described by the following MDP: The state space is $\{0, \dots, N\}$; the action space at $n$ is $\mathcal{A}(n) = \{0 < a \leq \min\{n, N - n\}\}$; the transition from state $n$ and action $a$ is $n - a$ and $n + a$ with probability $p$ and $1 - p$, respectively; the reward function is $r(N) = 1$ and $r(n) = 0$ for $0 \leq n \leq N - 1$. The MDP terminates at $n \in \{0, N\}$. We use a time-discount factor of $0 \leq \gamma \leq 1$, where the agent receives $\gamma^T r(N)$ rewards if the agents reaches the state $n = N$ at time $T$.

Let $z(n)$, $n \in \mathbb{N}, 0 \leq n \leq N$, be the value function. The exact solution below of the discrete case is relying on Theorem 12, our main theorem which describes the exact solution of the continuous case. This theorem will be discussed and proved later in Section A.1.

**Proposition 1.** *Let $0 \leq \gamma \leq 1$ and $p > 0.5$. The optimal value function $z(n)$ is $v(n/N)$ in the discrete setting of the Gambler's problem, where $v(\cdot)$ is the optimal value function under the continuous case defined in Theorem 12.*

*Proof.* We first verify the Bellman equation. By the definition of $v(\cdot)$ we have

$$z(n) = v(n/N)$$
$$= \max_{0 < a \leq \min\{n/N, 1 - n/N\}} p\gamma\, v(n/N - a) + (1 - p)\gamma\, v(n/N + a)$$
$$\geq \max_{0 < a \leq \min\{n/N, 1 - n/N\}, Na \in \mathbb{N}} p\gamma\, v(n/N - a) + (1 - p)\gamma\, v(n/N + a)$$
$$= \max_{0 < a \leq \min\{n, N - n\}, a \in \mathbb{N}} p\gamma\, z(n - a) + (1 - p)\gamma\, z(n + a).$$

Meanwhile let $a^* = \min\{n, N - n\}$, Corollary 13 suggests that

$$z(n) = v(n/N)$$
$$= p\gamma\, v((n - a^*)/N) + (1 - p)\gamma\, v((n + a^*)/N)$$
$$= p\gamma\, z(n - a^*) + (1 - p)\gamma\, v(n + a^*)$$
$$\leq \max_{0 < a \leq \min\{n, N - n\}, a \in \mathbb{N}} p\gamma\, z(n - a) + (1 - p)\gamma\, z(n + a).$$

Therefore $z(n) = \max_{0 < a \leq \min\{n, N - n\}, a \in \mathbb{N}} p\gamma\, z(n - a) + (1 - p)\gamma\, z(n + a)$ as desired.

We then show that $z(n) = v(n/N)$ is the unique function that satisfies the Bellman equation. The proof is similar to the proof of Lemma 2, but the arguments will be relatively easier, as both the state space and the action space are discrete. Let $f(n)$ also satisfy the Bellman equation. We desire to prove that $f(n)$ is identical to $z(n)$.

Define $\delta = \max_{1 \leq n \leq N - 1} f(n) - z(n)$. This maximum must exists as there are finite many states. Then define the non-empty set $S = \{n \mid f(n) - z(n) = \delta, 1 \leq n \leq N - 1\}$. For any $n' \in S$ and $a' \in \arg\max_{1 \leq n \leq \min\{n', N - n'\}} p\gamma\, f(n' - a) + (1 - p)\gamma\, f(n' + a)$, we have

$$f(n') = p\gamma\, f(n' - a') + (1 - p)\gamma\, f(n' + a')$$
$$\overset{(\heartsuit)}{\leq} p\gamma\, (z(n' - a') + \delta) + (1 - p)\gamma\, (z(n' + a') + \delta)$$
$$\leq p\gamma\, z(n' - a') + (1 - p)\gamma\, z(n' + a') + \delta$$
$$\leq z(n') + \delta$$
$$= f(n').$$

As the equality holds, by the equality of ($\heartsuit$) we have $n' - a' \in S$ and $n' + a' \in S$.

Now we specify some $n_0 \in S$ and $a_0 \in \arg\max_{1 \leq n \leq \min\{n_0, N - n_0\}} p\gamma\, f(n_0 - a) + (1 - p)\gamma\, f(n_0 + a)$. Then, we have $n_0 - a_0 \in S$. Denote $n_1 = n_0 - a_0$ and, recursively, $a_t \in \arg\max_{1 \leq n \leq \min\{n_t, N - n_t\}} p\gamma\, f(n_t - a) + (1 - p)\gamma\, f(n_t + a)$ and $n_{t+1} = n_t - a_t, t = 1, 2, \dots$; Since $a_t \geq 1$ and $n_t \in \mathbb{N}$, there must exist a $T$ such that $n_T = 0$. Therefore, $\delta = f(n_T) - z(n_T) = 0$.

By the same argument $\bar{\bar{\delta}} = \max_{1 \le n \le N-1} z(n) - f(n) = 0$. Therefore, $z(n)$ and $f(n)$ are identical, as desired.

As $z(n)$ is the unique function that satisfies the Bellman equation, it is the optimal value function of the problem. □

Proposition 1 indicates the discretization of the problems yields the discrete, exact evaluation of the continuous value function at $0, 1/N, \ldots, 1$. If we omit the learning error, the plots on the book and by the open source implementation (Zhang, 2019) are the evaluation of the fractal $v(s)$ at $0, 1/N, \ldots, 1$. This explains the strange appearance of the curve in the figures.

## 3 SETTING

We formulate the continuous Gambler's problem as a Markov decision process (MDP) with the state space $\mathcal{S} = [0, 1]$ and the action space $\mathcal{A}(s) = (0, \min\{s, 1-s\}], s \in (0, 1)$. Here $s \in \mathcal{S}$ represents the capital the gambler currently possesses and the action $a \in \mathcal{A}(s)$ denotes the amount of bet. Without loss of generality, we have assumed that the bet amount should be less or equal to $1 - s$ to avoid the total capital to be more than 1. The consecutive state $s'$ transits to $s - a$ and $s + a$ with probability $p \ge 0.5$ and $1 - p$ respectively. The process terminates if $s \in \{0, 1\}$ and the agent receives an episodic reward $r = s$ at the terminal state. Let $0 \le \gamma \le 1$ be the discount factor.

Let $f : [0, 1] \to \mathbb{R}$ be a real function. For $f(s)$ to be the optimal value function, the Bellman equation for the non-terminal and terminal states are

$$f(s) = \max_{a \in \mathcal{A}(s)} p\gamma\, f(s - a) + (1 - p)\gamma\, f(s + a) \ \text{ for any } \ s \in (0, 1), \tag{A}$$

and

$$f(0) = 0, \ \ f(1) = 1. \tag{B}$$

It can be shown (later in Lemma 2 and Lemma 3) that a function satisfying (AB) must be lower bounded by 0. A reasonable upper bound is 1, as the value function is the probability of the gambler eventually reaching the target, which must be between 0 and 1. It is also reasonable to assume the continuity of the value function at $s = 0$. Otherwise an arbitrary small amount of starting capital will have at least a constant probability of reaching the target 1.[1] Consequently the expectation of capital at the end of the game is greater than the starting capital, which contradicts $p \ge 0.5$. The bounded version (X) of the problem leads to the optimal value function.

$$0 \le \gamma \le 1, \ p > 0.5, \ f(s) \le 1 \text{ for all } s, \ f(s) \text{ is continuous on } s = 0. \tag{X}$$

Respectively, the unbounded version (Y) of the problem leads to the solutions of the Bellman equation.

$$0 \le \gamma \le 1, \ p > 0.5. \tag{Y}$$

The results extend for $p = 0.5$ in general, except an extreme corner case of $\gamma = 1$, $p = 0.5$, where the monotonicity in Lemma 3 will not apply. This case (Z) involves arguments over measurability and the belief of Axiom of Choice, which we will discuss at the end of Section A.

$$\gamma = 1, \ p = 0.5, \ f(s) \text{ is unbounded.} \tag{Z}$$

We are mostly interested in two settings: the first setting (ABX) and its solution Theorem 12, discuss a set of necessary conditions of $f(s)$ being the optimal value function of the Gambler's problem. As we show later the solution of (ABX) is unique, this solution must be the optimal value function. The second setting (ABY) and its solutions in Proposition 21 and Theorem 22 discuss all the functions that satisfy the Bellman equation. These functions are the optimal points that value iteration and Q-learning algorithms may converge to. (ABZ) is interestingly connected some foundations of mathematics like the belief of axioms, and is discussed in Theorem 27.

---

[1]This continuity assumption is only for a better organization of the settings. The more general problem (AB) is solved in Section A.2 without this assumption.

## 4 ANALYSIS

The analysis section rigorously supports the statements on the Gambler's problem and its Bellman equation with proofs and discussions. It is deferred to the appendix due to the page limit.

## 5 CONCLUSION AND FUTURE WORKS

We give a complete solution to the Gambler's problem, a simple and classic problem in reinforcement learning, under a variety of settings. We show that its optimal value function is very complicated and even pathological. Despite its seeming simpleness, these results are not clearly pointed out in previous studies.

Our contributions are the theoretical finding and the implications. It is worthy to bring the current results to start the discussion among the community. Indicated by the Gambler's problem, the current algorithmic approaches in reinforcement learning might underestimate the complexity. We expect more evidence could be found in the future and new algorithms and implementations could be brought out.

It would be interesting to see how these results of the Gambler's problem generalized to other MDPs. Finding these characterizations of MDPs is in general an important step to understand reinforcement learning and sequential decision processes.

## ACKNOWLEDGEMENT

We thank Richard S. Sutton and Andrew Barto for raising the Gambler's problem in their book, and Rich S. Sutton for the discussions on our theorems and implications. We thank Andrej Bogdanov for pointing out the connection to the Axiom of Choice, namely, Theorem 27, and Chengyu Lin for the discussions on the properties of $v(s)$, namely, Lemma 6, 8, and 9. This paper was largely improved by the reviews and comments. We especially would like to thank Csaba Szepesvári, Kirby Banman, and ICLR 2020 anonymous reviewers for their helpful feedback.

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

## A ANALYSIS

### A.1 ANALYSIS OF THE GAMBLER'S PROBLEM

In this section we show that $v(s)$ defined below is a unique solution of the system (ABX). Since the optimal state-value function must satisfies the system (ABX), $v(s)$ is the optimal state-value function of the Gambler's problem. This statement is rigorously proved in Theorem 12.

Let $0 \leq \gamma \leq 1$ and $p > 0.5$. We define $v(1) = 1$, and

$$v(s) = \sum_{i=1}^{\infty} (1-p)\gamma^i b_i \prod_{j=1}^{i-1} ((1-p) + (2p-1)b_j) \tag{1}$$

for $0 \leq s < 1$, where $s = 0.b_1 b_2 \ldots b_\ell \ldots_{(2)}$ is the binary representation of $s$. It is obvious that the series converges for any $0 \leq s < 1$.

The notation $v(s)$ will always refer to the definition above in this paper and will not change with the context. We show later that this $v(s)$ is the optimal value function of the problem. We use the notation $f(s)$ to denote a general solution of a system, which varies according to the required properties.

Let the set of dyadic rationals

$$G_\ell = \left\{ k2^{-\ell} \mid k \in \left\{ 1, \ldots, 2^\ell - 1 \right\} \right\} \tag{3}$$

such that $G_\ell$ is the set of numbers that can be represented by at most $\ell$ binary bits. The general idea to verify the Bellman equation (AB) is to prove

$$v(s) = \max_{a \in G_\ell \cap \mathcal{A}(s)} (1-p)\gamma \, v(s+a) + p\gamma \, v(s-a) \text{ for any } s \in G_\ell$$

by induction of $\ell = 1, 2, \ldots$, and generalize this optimality to the entire interval $s \in (0, 1)$.

It then suffices to show the uniqueness of $v(s)$ that solves the system (ABX). This is proved by assuming the existence of a solution $f(s)$ and derive the identity $f(s) = v(s)$, condition on the Bellman property that $v(s)$ satisfies (AB). For presentation purposes, the uniqueness is discussed first.

As an overview, Lemma 2, 3, and 4 describe the system (ABX). Claim 5, Lemma 6, 8, and 9 describe the properties of $v(s)$.

**Lemma 2** (Uniqueness under existence). *Let $f(s) : [0, 1] \to \mathbb{R}$ be a real function. If $v(s)$ and $f(s)$ both satisfy (ABX), then $v(s) = f(s)$ for all $0 \leq s \leq 1$.*

*Proof.* We proof the lemma by contradiction. Assume that $f(s)$ is also a solution of the system such that $f(s)$ is not identical with $v(s)$ at some $s$. Define $\delta = \sup_{0<s<1} f(s) - v(s)$. As $f(2^{-1}) \geq (1-p)\gamma \, f(1) + p\gamma \, f(0) = (1-p)\gamma = v(2^{-1})$, we have $\delta \geq 0$.

We show that $\delta$ cannot be zero by contradiction. If $\delta$ is zero, as $v(s)$ and $f(s)$ are not identical, there exists an $s$ such that $f(s) < v(s)$. In this case, let $\bar{\delta} = \sup_{0<s<1} v(s) - f(s)$. Then we choose $\bar{\epsilon} = (1-p\gamma)\bar{\delta}$ and specify $s_0$ such that $v(s_0) - f(s_0) > \bar{\delta} - \bar{\epsilon}$. Let $a_0 = \min\{s_0, 1-s_0\}$, we have

$$\begin{aligned}
v(s_0) &= (1-p)\gamma \, v(s_0 - a_0) + p\gamma \, v(s_0 + a_0) \\
&\leq (1-p)\gamma \, f(s_0 - a_0) + p\gamma \, f(s_0 + a_0) + p\gamma\bar{\delta} \\
&\leq f(s_0) + p\gamma\bar{\delta}.
\end{aligned}$$

The above inequality is due to at least one of $s_0 - a_0 = 0$ and $s_0 + a_0 = 1$ must hold. Thus at least one of $v(s_0 - a_0) - f(s_0 - a_0)$ and $v(s_0 + a_0) - f(s_0 + a_0)$ must be zero. The inequality contradicts $v(s_0) - f(s_0) > \bar{\delta} - \bar{\epsilon}$. Hence, $\delta$ cannot be zero. We discuss under $\delta > 0$ for the rest of the proof.

**Case (I): $\gamma < 1$.** In this case, we choose $\epsilon = (1 - \gamma)\delta$. By the definition of $\delta$ we specify $s_0$ such that $f(s_0) > v(s_0) + \delta - \epsilon$. In fact, the existence of $s_0$ is by the condition $\gamma < 1$. Let $a_0 \in \arg\max_{a \in \mathcal{A}(s_0)} p\gamma\, f(s_0 - a) + (1 - p)\gamma\, f(s_0 + a)$, we have

$$
\begin{aligned}
f(s_0) &= p\gamma\, f(s_0 - a_0) + (1 - p)\gamma\, f(s_0 + a_0) \\
&\le p\gamma\, (v(s_0 - a_0) + \delta) + (1 - p)\gamma\, (v(s_0 + a_0) + \delta) \\
&= p\gamma\, v(s_0 - a_0) + (1 - p)\gamma\, v(s_0 + a_0) + \gamma\delta \\
&\le v(s_0) + \delta - \epsilon.
\end{aligned}
$$

The inequality $f(s_0) \le v(s_0) + \delta - \epsilon$ contradicts $f(s_0) > v(s_0) + \delta - \epsilon$. Hence, the lemma is proved for the case $\gamma < 1$.

**Case (II): $\gamma = 1$.** When there exists an $s'$ such that $f(s') - v(s') = \delta$, we show the contradiction. Let $S = \{s | f(s) - v(s) = \delta, 0 < s < 1\} \ne \varnothing$. For any $s' \in S$ and $a' \in \arg\max_{a \in \mathcal{A}(s')} p\gamma\, f(s' - a) + (1 - p)\gamma\, f(s' + a)$, we have

$$
\begin{aligned}
f(s') &= p\gamma\, f(s' - a') + (1 - p)\gamma\, f(s' + a') \\
&= p\, f(s' - a') + (1 - p)\, f(s' + a') \\
&\overset{(\heartsuit)}{\le} p\, (v(s' - a') + \delta) + (1 - p)\, (v(s' + a') + \delta) \\
&= p\, v(s' - a') + (1 - p)\, v(s' + a') + \delta \\
&\overset{(\blacklozenge)}{\le} v(s') + \delta.
\end{aligned}
$$

Thus, the equality in $(\heartsuit)$ and $(\blacklozenge)$ must hold. We specify $s_0 \in S$, and by the equality of $(\heartsuit)$ we have $f(s_0 - a_0) = v(s_0 - a_0) + \delta$, thus $s_0 - a_0 \in S$. Let $s_1 = s_0 - a_0$, and we recursively specify an arbitrary $a_t \in \arg\max_{a \in \mathcal{A}(s_t)} (1 - p)\gamma\, f(s_t + a) + p\gamma\, f(s_t - a)$ and $s_{t+1} = s_t - a_t$, for $t = 1, 2, \ldots$, until $s_T = 0$ for some $T$, or indefinitely if such an $s_T$ does not exist. If $s_T$ exists and the sequence $\{s_t\}$ terminates at $s_T = 0$, then $f(s_T) = v(s_T) + \delta = \delta$ by $(\heartsuit)$, which contradicts the boundary condition $f(s_T) = f(0) = 0$.

We desire to show the existence of $T$. When there exists $t$ and $\ell$ such that $s_t \in G_\ell$, by Corollary 7 we have $s_{t+1} \in G_\ell$ and inductively $s_{t'} \in G_\ell$ for all $t' \ge t$. Consider that $\{s_t\}$ is strictly decreasing and there are finite many elements in $G_\ell$, $\{s_t\}$ cannot be infinite. Otherwise $s_t \notin G_\ell$ for any $t, \ell \ge 1$. Then by Corollary 10 the uniqueness of optimal action we have $s_{t+1} = 2s_t - 1$ if $s_t \ge \frac{1}{2}$, and $s_{t+1} = 0$ if $s_t \le \frac{1}{2}$. After finite many steps of $s_{t+1} = 2s_t - 1$ we will have $s_t = 0$ for some $t$.

It amounts to show the existence of $s'$ such that $f(s') - v(s') = \delta$. By Lemma 9 we have the continuity of $v(s)$. Lemma 3 indicates the monotonicity of $f(s)$ on $[0, 1)$. The upper bound $f(s) \le f(1)$ in (X) extends this monotonicity to the closed interval $[0, 1]$. Then by Lemma 4 we have the continuity of $f(s)$ on $(0, 1]$. By (X) this extends to $[0, 1]$. Thus we have the continuity of $f(s) - v(s)$, and consequently the existence of $\max_{0 \le s' \le 1} f(s') - v(s')$. As $f(0) - v(0) = f(1) - v(1) = 0$ and $\delta > 0$, this maximum must be attained at some $s' \in (0, 1)$. Therefore we have the existence of $\max_{0 < s' < 1} f(s') - v(s')$, which concludes the lemma. $\qquad\square$

**Lemma 3** (Monotonicity). *Let $\gamma = 1$ and $p > 0.5$. If a real function $f(s)$ satisfies (AB) then $f(s)$ is monotonically increasing on $[0, 1)$.*

*Proof.* We prove the claim by contradiction. Assume that there exists $s_1 < s_2$ where $f(s_1) > f(s_2)$. Denote $\Delta s = s_2 - s_1 > 0$ and $\Delta f = f(s_1) - f(s_2) > 0$. By induction we have

$$
f(s_2 - 2^{-\ell}\Delta s) - f(s_2) \ge p^\ell \Delta f
$$

for an arbitrary integer $\ell \ge 1$. Then when $s_2 + 2^{-\ell}\Delta s < 1$, by $f(s_2) \ge pf(s_2 - 2^{-\ell}\Delta s) + (1 - p)f(s_2 + 2^{-\ell}\Delta s)$,

$$
f(s_2 + 2^{-\ell}\Delta s) \le \frac{1}{1 - p}f(s_2) - \frac{p}{1 - p}f(s_2 - 2^{-\ell}\Delta s)
$$

$$= f(s_2) + \frac{p}{1-p}(f(s_2) - f(s_2 - 2^{-\ell}\Delta s))$$

$$\le f(s_2) + f(s_2) - f(s_2 - 2^{-\ell}\Delta s).$$

This concludes $f(s_2 + 2^{-\ell}\Delta s) - f(s_2) \le f(s_2) - f(s_2 - 2^{-\ell}\Delta s)$. By induction we have

$$f(s_2 + k2^{-\ell}\Delta s) - f(s_2 + (k-1)2^{-\ell}\Delta s) \le f(s_2 + (k-1)2^{-\ell}\Delta s) - f(s_2 - (k-2)2^{-\ell}\Delta s)$$

for $k = 1, 2, \ldots$, when $s_2 + k2^{-\ell}\Delta s < 1$. We sum this inequality over $k$ and get

$$f(s_2 + k2^{-\ell}\Delta s) - f(s_2) \le k(f(s_2) - f(s_2 - 2^{-\ell}\Delta s))$$

$$\le -kp^\ell\Delta f.$$

By letting $k = 2^n$, $\ell = n + n_0$, $s_2 + 2^{-n_0}\Delta s < 1$, and $n \to +\infty$, we have $s_2 + k2^{-\ell}\Delta s < 1$ and $-kp^\ell\Delta f \to -\infty$. The arbitrariness of $n$ indicates the non-existence of $f(s_2 + k2^{-\ell}\Delta s)$, which contradicts the existence of the solution $f(\cdot)$. $\qquad\square$

**Lemma 4** (Continuity). *Let $\gamma = 1$ and $p \ge 0.5$. If a real function $f(s)$ is monotonically increasing on $(0, 1]$ and it satisfies (AB), then $f(s)$ is continuous on $(0, 1]$.*

*Proof.* We show the continuity by contradiction. Suppose that there exists a point $s' \in (0, 1)$ such that $f(s)$ is discontinuous at $s'$, then there exists $\epsilon, \delta > 0$ where $f(s' + \epsilon_1) - f(s' - \epsilon_2) \ge \delta$ for any $\epsilon_1 + \epsilon_2 = \epsilon$. Then, by

$$f(s' - \frac{1}{4}\epsilon) \ge p\, f(s' - \epsilon) + (1 - p)\, f(s' + \frac{2}{4}\epsilon),$$

we have

$$f(s' - \frac{1}{4}\epsilon) - f(s' - \epsilon) \ge (1 - p)\delta/p.$$

Similarly, for $k = 1, 2, \ldots,$

$$f(s' - \frac{1}{4^k}\epsilon) - f(s' - \frac{1}{4^{k-1}}\epsilon) \ge (1 - p)\delta/p.$$

Let $k > ((1 - p)\delta/p)^{-1}$, we have $f(s' - \frac{1}{4^k}\epsilon) - f(s' - \epsilon) \ge 1$. This contradict with the fact that $f(s)$ is bounded between $0$ and $1$. The continuity follows on $(0, 1)$.

If the function is discontinuous on $s = 1$, then there exists $\epsilon, \delta > 0$ where $f(1) - f(1 - \epsilon_1) \ge \delta$ for any $\epsilon_1 \le \epsilon$. The same argument holds by observing

$$f(1 - \frac{1}{2^{k-1}}\epsilon) \ge p\, f(1 - \frac{1}{2^k}\epsilon) \ge f(1).$$

The lemma follows. $\qquad\square$

When $f(s)$ is only required to be monotonically increasing on $(0, 1)$, the continuity still holds but only on $(0, 1)$.

For simplicity define

$$Q_v(s, a) = p\gamma\, v(s - a) + (1 - p)\gamma\, v(s + a). \tag{4}$$

As $v(s)$ is the optimal state-value function (to be proved later in Theorem 12), $Q_v(s, a)$ is in fact the optimal action-value function (Sutton & Barto, 2018; Szepesvári, 2010).

Recall that $G_\ell$ is the set of dyadic rationals $\{k2^{-\ell} \mid k \in \{1, \ldots, 2^\ell - 1\}\}$.

**Claim 5.** *For any $s = 0.b_1b_2\ldots b_{\ell(2)} \in G_\ell \cup \{0\}$,*

$$v\left(s + 2^{-(\ell+1)}\right) - v(s) = (1 - p)\gamma^{\ell+1}\prod_{j=1}^{\ell}((1 - p) + (2p - 1)b_j) \le (1 - p)p^\ell\gamma^{\ell+1}. \tag{5}$$

*For any $s = 0.b_1 b_2 \ldots b_{k\,(2)} \in G_\ell$ with $b_k = 1$ and $1 \le k \le \ell$,*

$$v(s) - v\left(s - 2^{-(\ell+1)}\right) \ge p^{\ell-k+1}(1-p)\gamma^{\ell+1} \prod_{j=1}^{k-1} \left((1-p) + (2p-1)b_j\right). \tag{6}$$

*Also,*

$$v(1) - v\left(1 - 2^{-(\ell+1)}\right) \ge p^{\ell+1}\gamma^{\ell+1}. \tag{7}$$

*The equality of (6) and (7) holds if and only if $\gamma = 1$.*

*Proof.* Inequality (5) and (7) are obtained by the definition of $v(s)$. To derive inequality (6), denote $k = \max\{1 \le i \le \ell : b_i = 0\}$ and then

$$
\begin{aligned}
&v(s) - v\left(s - 2^{-(\ell+1)}\right) \\
&= (1-p)\gamma^k \prod_{j=1}^{k-1}\left((1-p) + (2p-1)b_j\right) \\
&\quad - \sum_{i=k+1}^{\ell+1}(1-p)\gamma^i \cdot 1 \cdot \prod_{j=1}^{k-1}\left((1-p) + (2p-1)b_j\right) \cdot (1-p) \cdot \prod_{j=k+1}^{i-1} p \\
&= (1-p)\gamma^k \prod_{j=1}^{k-1}\left((1-p) + (2p-1)b_j\right) \cdot \left(1 - (1-p)\sum_{i=k+1}^{\ell+1}\gamma^{i-k}p^{i-k-1}\right) \\
&= (1-p)\gamma^k \prod_{j=1}^{k-1}\left((1-p) + (2p-1)b_j\right) \cdot \left(1 - (1-p)\gamma\frac{1-(\gamma p)^{\ell-k+1}}{1-\gamma p}\right) \\
&\ge (1-p)\gamma^k \prod_{j=1}^{k-1}\left((1-p) + (2p-1)b_j\right) \cdot \left(1 - \left(1-(\gamma p)^{\ell-k+1}\right)\right) \\
&= (1-p)p^{\ell-k+1}\gamma^{\ell+1}\prod_{j=1}^{k-1}\left((1-p) + (2p-1)b_j\right).
\end{aligned}
$$

The arguments are due to the fact that $s - 2^{-(\ell+1)} = 0.b_1 b_2 \ldots b_{k-1} 0_k 1_{k+1} \ldots 1_{\ell+1\,(2)}$ and the inequality is by $(1-p)\gamma \le 1 - \gamma p$. □

**Lemma 6.** *Let $\ell \ge 1, 0 < \gamma \le 1, 0.5 \le p < 1$. For any $s \in G_\ell$,*

$$\max_{a \in (G_{\ell+1} \backslash G_\ell) \cap \mathcal{A}(s)} Q_v(s,a) \le \max_{a \in G_\ell \cap \mathcal{A}(s)} Q_v(s,a).$$

*Proof.* **Case (I):** First we prove that for $\ell > 1$, any $s \in G_\ell, a \in G_\ell \cap \mathcal{A}(s), a > 2^{-\ell}$, and $s + a < 1$,

$$Q_v\left(s, a - 2^{-(\ell+1)}\right) \le \max\left\{Q_v(s,a), Q_v\left(s, a - 2^{-\ell}\right)\right\}.$$

Note that in this case, $a - 2^{-(\ell+1)} \in G_{\ell+1} \cap \mathcal{A}(s)$ and $a - 2^{-\ell} \in G_\ell \cap \mathcal{A}(s)$.

Let $s - a = 0.c_1 c_2 \ldots c_{\ell\,(2)} = 0.c_1 c_2 \ldots 0_k 1_{k+1} \ldots 1_{\ell\,(2)}$, where $k = \max\{1 \le i \le \ell : c_i = 0\}$ is the index of the last 0 bit in $s - a$. Such $k$ must exist since $0 \le s - a \le 1 - 3 \times 2^{-\ell} < 1$. Similarly, let $s + a = 0.d_1 d_2 \ldots d_{\ell\,(2)} = 0.d_1 d_2 \ldots 1_{k'\,(2)}$ where $k' = \max\{1 \le i \le \ell : d_i = 1\}$ is the index of the last 1 bit in $s + a$. Such $k'$ must exist since $3 \times 2^{-\ell} \le s + a < 1$. Also, $s + a - 2^{-(\ell+1)} = 0.d_1 d_2 \ldots 0_{k'} 1_{k'+1} \ldots 1_{\ell+1\,(2)}$.

To prove $Q_v\left(s, a - 2^{-(\ell+1)}\right) \le Q_v(s,a)$, it is equivalent to prove

$$v(s+a) - v\left(s+a-2^{-(\ell+1)}\right) \ge \frac{p}{1-p}\left(v\left(s-a+2^{-(\ell+1)}\right) - v(s-a)\right).$$

Then by applying inequality (6), (7) and inequality (5) in Claim 5 on the LHS and RHS respectively, it suffices to prove

$$p^{\ell-k'}(1-p)\prod_{j=1}^{k'-1}((1-p)+(2p-1)d_j) \geq \prod_{j=1}^{\ell}((1-p)+(2p-1)c_j)$$

$$= p^{\ell-k}(1-p)\prod_{j=1}^{k-1}((1-p)+(2p-1)c_j).$$

Let $M_c = c_1 + \cdots + c_{k-1}, M_d = d_1 + \cdots + d_{k'-1}$ be the number of 1s in $\{c_1, \ldots, c_k\}, \{d_1, \ldots, d_{k'}\}$ respectively. Then $Q_v\left(s, a - 2^{-(\ell+1)}\right) \leq Q_v(s, a)$ holds when $p = 0.5$ or $p > 0.5, M_c + k \geq M_b + k'$.

To prove $Q_v\left(s, a - 2^{-(\ell+1)}\right) \leq Q_v\left(s, a - 2^{-\ell}\right)$, it is equivalent to prove

$$v\left(s - a + 2^{-\ell}\right) - v\left(s - a + 2^{-(\ell+1)}\right) \geq \frac{1-p}{p}\left(v\left(s + a - 2^{-(\ell+1)}\right) - v\left(s + a - 2^{-\ell}\right)\right).$$

Note that $s - a + 2^{-\ell} = 0.c_1c_2\ldots 1_{k(2)}$ and $s + a - 2^{-\ell} = 0.d_1d_2\ldots 0_{k'}1_{k'+1}\ldots 1_{\ell(2)}$. Then by inequality (6), (7) and inequality (5) on the LHS and RHS respectively, it suffices to prove

$$p^{\ell-k+2}\prod_{j=1}^{k-1}((1-p)+(2p-1)c_j) \geq (1-p)^2 p^{\ell-k'}\prod_{j=1}^{k'-1}((1-p)+(2p-1)d_j).$$

Then $Q_v\left(s, a - 2^{-(\ell+1)}\right) \leq Q_v\left(s, a - 2^{-\ell}\right)$ holds when $p = 0.5$ or $p > 0.5, M_c + k' + 2 \geq M_d + k$.

As at least one of $M_c + k' + 1 \geq M_d + k$ and $M_d + k \geq M_c + k' + 1$ holds, thus $Q_v\left(s, a - 2^{-(\ell+1)}\right) < \max\left\{Q_v(s, a), Q_v\left(s, a - 2^{-\ell}\right)\right\}$.

We cover two corner cases for the completeness of the proof.

**Case (II):** Next we prove for $\ell \geq 1$, any $s \in G_\ell, a \in G_\ell \cap \mathcal{A}(s)$ and $s + a = 1$,

$$Q_v\left(s, a - 2^{-(\ell+1)}\right) \leq Q_v(s, a).$$

Similar to above, it is equivalent prove

$$v(1) - v\left(1 - 2^{-(\ell+1)}\right) \geq \frac{p}{1-p}\left(v\left(s - a + 2^{-(\ell+1)}\right) - v(s-a)\right).$$

Note that by Claim 5,

$$v(1) - v\left(1 - 2^{-(\ell+1)}\right) \geq p^{\ell+1}\gamma^{\ell+1} = \frac{p}{1-p} \cdot (1-p)p^\ell\gamma^{\ell+1}$$

$$\geq \frac{p}{1-p}\left(v\left(s - a + 2^{-(\ell+1)}\right) - v(s-a)\right),$$

which concludes the proof.

**Case (III):** Last we prove for $\ell > 1$, any $s \in G_\ell$ and $a = 2^{-\ell}, s < 1 - 2^{-\ell}$.

When $s = 0.b_1b_2\ldots 0_m 1_{m+1}\ldots 1_{\ell(2)}$ with $1 \leq m < \ell$, to prove $Q_v\left(s, 2^{-(\ell+1)}\right) \leq Q_v\left(s, 2^{-\ell}\right)$, it is equivalent to prove

$$v\left(s + 2^{-\ell}\right) - v\left(s + 2^{-(\ell+1)}\right) \geq \frac{p}{1-p}\left(v\left(s - a + 2^{-(\ell+1)}\right) - v(s-a)\right).$$

In this case, $s + 2^{-\ell} = 0.b_1b_2\ldots 1_{m(2)}, s - 2^{-\ell} = 0.b_1b_2\ldots 0_m 1_{m+1}\ldots 1_{\ell-1(2)}$ and $M_c = M_d, k = k' = m$, thus $M_d + k \geq M_c + k'$, which concludes the proof similar to the first part of the (I) case.

When $s = 0.b_1b_2\ldots 1_{m'}0_{m'+1}\ldots 0_{\ell(2)}$ with $1 \leq m' < \ell$, let $M_b = b_1 + \ldots b_{m'}$. Then,

$$Q_v\left(s, 2^{-m'}\right) - Q_v\left(s, 2^{-(\ell+1)}\right)$$

$$= (1-p)\gamma(v(s+2^{-m'}) - v(s-2^{-m'})) - (1-p)\gamma(v(s) - v(s-2^{-m'}))$$
$$\quad - (1-p)\gamma(v(s+2^{-\ell}) - v(s)) - p\gamma(v(s-2^{-\ell}) - v(s-2^{-m'}))$$
$$\geq (1-p)\gamma(p\gamma)^{M_b}((1-p)\gamma)^{m'-2-M_b}(1-p)\gamma - (1-p)\gamma(p\gamma)^{M_b}((1-p)\gamma)^{m'-1-M_b}(1-p)\gamma$$
$$\quad - (1-p)\gamma(p\gamma)^{M_b+1}((1-p)\gamma)^{\ell-2-M_b}(1-p)\gamma$$
$$\quad - p\gamma(p\gamma)^{M_b}((1-p)\gamma)^{m'-M_b}(1 + (p\gamma) + \dots (p\gamma)^{\ell-m'-1})(1-p)\gamma$$
$$= p^{M_b}(1-p)^{m'-M_b}\gamma^{m'} - p^{M_b}(1-p)^{m'+1-M_b}\gamma^{m'+1} - p^{M_b+1}(1-p)^{\ell-M_b}\gamma^{\ell+1}$$
$$\quad - p^{M_b+1}(1-p)^{m'+1-M_b}\gamma^{m'+2}(1 - (p\gamma)^{\ell-m'})/(1-p\gamma)$$
$$\geq p^{M_b}(1-p)^{m'-M_b}\gamma^{m'} - p^{M_b}(1-p)^{m'+1-M_b}\gamma^{m'+1} - p^{M_b+1}(1-p)^{\ell-M_b}\gamma^{\ell+1}$$
$$\quad - p^{M_b+1}(1-p)^{m'-M_b}\gamma^{m'+1}(1 - (p\gamma)^{\ell-m'})$$
$$\geq -p^{M_b+1}(1-p)^{\ell-M_b}\gamma^{\ell+1} - p^{M_b+1}(1-p)^{m'-M_b}\gamma^{m'+1}(-(p\gamma)^{\ell-m'})$$
$$\geq 0. \hspace{8cm} \square$$

The arguments in the proof that either $M_c + k \geq M_d + k' + 1$ or $M_d + k' \geq M_c + k$ must hold is tight for integers $M_c$ and $M_d$. This is the case for $a \in G_{\ell+1} \setminus G_\ell$. When $a \notin G_{\ell+1}$, this sufficient condition becomes even looser. The lemma imposes $G_\ell$ to be the only set of possible optimal actions, given $s \in G_\ell$.

**Corollary 7.** *Let $\ell \geq 1$. For any $s \in G_\ell$,*
$$\underset{a \in \mathcal{A}(s)}{\operatorname{argmax}} \; Q_v(s,a) \subseteq G_\ell.$$

Now we verify the Bellman property on $\bigcup_{\ell \geq 1} G_\ell$.

**Lemma 8.** *Let $\ell \geq 1$. For any $s \in G_{\ell+1}$,*
$$\min\{s, 1-s\} \in \underset{a \in G_{\ell+1} \cap \mathcal{A}(s)}{\operatorname{argmax}} \; Q_v(s,a).$$

*Proof.* We prove the lemma by induction over $\ell$. When $\ell = 1$, it is obvious since $G_1$ has only one element. The base case $\ell = 2$ is also immediate by exhausting $a \in \{2^{-1}, 2^{-2}\}$ for $s = 2^{-1}$. Now we assume that for any $s \in G_\ell$, $\min\{s, 1-s\} \in \operatorname{argmax}_{a \in G_\ell \cap \mathcal{A}(s)} Q_v(s,a)$. We aim to prove this lemma for $\ell + 1$.

For $s \in G_\ell$, by Lemma 6, $\operatorname{argmax}_{a \in G_{\ell+1} \cap \mathcal{A}(s)} Q_v(s,a) \subseteq G_\ell$. Then by the induction assumption, $\min\{s, 1-s\} \in \operatorname{argmax}_{a \in G_\ell \cap \mathcal{A}(s)} Q_v(s,a) \subseteq \operatorname{argmax}_{a \in G_{\ell+1} \cap \mathcal{A}(s)} Q_v(s,a)$. Hence, the lemma holds for $s \in G_\ell$. We discuss under $s \in G_{\ell+1} \setminus G_\ell$ for the rest of the proof.

We start with two inductive properties of $v(s)$ to reduce the problem from $s \in G_{\ell+1}$ to $s' \in G_\ell$, where $s'$ is either $2s$ or $2s - 1$. For any $s \geq 2^{-1}$, that is, $s = 0.c_1 c_2 \dots c_{\ell+1 (2)} \in G_{\ell+1}$ with $c_1 = 1$,

$$v(s) = \sum_{i=1}^{\ell}(1-p)\gamma^i c_i \prod_{j=1}^{i-1}((1-p) + (2p-1)c_j)$$
$$= (1-p)\gamma + \sum_{i=2}^{\ell}(1-p)\gamma^i c_i \prod_{j=1}^{i-1}((1-p) + (2p-1)c_j)$$
$$= (1-p)\gamma + \sum_{i=1}^{\ell-1}(1-p)\gamma^{i+1} c_{i+1}((1-p) + (2p-1)c_1) \prod_{j=1}^{i-1}((1-p) + (2p-1)c_{j+1})$$
$$= (1-p)\gamma + p\gamma \, v(0.c_2 \dots c_{\ell+1 (2)})$$
$$= (1-p)\gamma + p\gamma \, v(2s - 1).$$

Similarly, for any $s < 2^{-1}$, that is, $s = 0.c_1 c_2 \dots c_{\ell+1 (2)} \in G_{\ell+1}$ with $c_1 = 0$,

$$v(s) = \sum_{i=1}^{\ell-1}(1-p)^2 \gamma^{i+1} c_{i+1} \prod_{j=1}^{i-1}((1-p) + (2p-1)c_{j+1})$$

$$= (1-p)\gamma v(2s).$$

Armed with the properties, we split the discussion into four cases $2^{-1} + 2^{-2} \le s < 1$, $2^{-1} \le s < 2^{-1} + 2^{-2}$, $2^{-1} - 2^{-2} < s < 2^{-1}$, and $0 < s \le 2^{-1} - 2^{-2}$.

When $s \ge 2^{-1} + 2^{-2}$, As $a \le 1 - s$, we have $s - a \ge 2^{-1}$ and $s + a \ge 2^{-1}$. Hence, the first bit after the decimal of $s - a$ and $s + a$ is 1. Hence,

$$
\begin{aligned}
Q_v(s, a) &= p\gamma\, v(s - a) + (1 - p)\gamma\, v(s + a) \\
&= (1 - p)\gamma^2 + p\gamma\, (p\gamma\, v(2s - 2a - 1)) + (1 - p)\gamma\, v(2s + 2a - 1) \\
&= (1 - p)\gamma^2 + p\gamma\, (p\gamma\, v((2s - 1) - 2a) + (1 - p)\gamma\, v((2s - 1) + 2a)) \\
&= (1 - p)\gamma^2 + p\gamma\, Q_v(2s - 1, 2a).
\end{aligned}
$$

As $2s - 1 \in G_\ell$ and $2a \in G_\ell$, by the induction assumption the maximum of $Q_v(2s - 1, 2a)$ is obtained at $a = 1 - s$. Hence, $1 - s \in \mathrm{argmax}_{a \in G_{\ell+1} \cap \mathcal{A}(s)} Q_v(s, a)$ as desired.

When $2^{-1} \le s < 2^{-1} + 2^{-2}$, if $s - a \ge 2^{-1}$, then first bit after the decimal of $s - a$ and $s + a$ is 1 and the lemma follows the same arguments as the above case. Otherwise, if $s - a < 2^{-1}$, we have

$$
\begin{aligned}
Q_v(s, a) &= p\gamma\, v(s - a) + (1 - p)\gamma\, v(s + a) \\
&= (1 - p)^2\gamma^2 + p(1 - p)\gamma^2\, v(2s - 2a) + p(1 - p)\gamma^2\, v(2s + 2a - 1) \\
&= (1 - p)\gamma\, (p\gamma\, v((2s - 2^{-1}) - (2a - 2^{-1})) + (1 - p)\gamma\, v((2s - 2^{-1}) + (2a - 2^{-1}))) \\
&\quad + (1 - p)(2p - 1)\gamma^2\, v(2s + 2a - 1) + (1 - p)^2\gamma^2 \\
&= (1 - p)\gamma\, Q_v(2s - 2^{-1}, 2a - 2^{-1}) + (1 - p)(2p - 1)\gamma^2\, v(2s + 2a - 1) + (1 - p)^2\gamma^2.
\end{aligned}
$$

As $2s - 2^{-1} \in G_\ell$ and $2a - 2^{-1} \in G_\ell$ whenever $l \ge 2$, by the induction assumption $Q_v(2s - 2^{-1}, 2a - 2^{-1})$ obtains its maximum at $a = 1 - s$. By Claim 5, $v(s)$ is monotonically increasing on $G_\ell$ for any $\ell \ge 2$. Hence, $v(2s + 2a - 1)$ obtains the maximum at the maximum feasible $a$, which is $a = 1 - s$. Since both terms takes their respective maximum at $a = 1 - s$, we conclude that $1 - s \in \mathrm{argmax}_{a \in G_{\ell+1} \cap \mathcal{A}(s)} Q_v(s, a)$ as desired.

The other two cases, $2^{-1} - 2^{-2} < s < 2^{-1}$ and $0 < s \le 2^{-1} - 2^{-2}$, follow similar arguments. The lemma follows. $\qquad\square$

**Lemma 9.** *Both $v(s)$ and $v'(s) = \max_{a \in \mathcal{A}(s)} Q_v(s, a)$ are continuous at $s$ if there does not exist an $\ell$ such that $s \in G_\ell$.*

*Proof.* We first proof the continuity of $v(s)$. For $s = b_1 b_2 \ldots b_\ell \ldots_{(2)}$, $s \notin G_\ell$ indicates that for any integer $N$ there exists $n_1 \ge N$ such that $b_{n_1} = 1$ and $n_0 \ge N$ such that $b_{n_0} = 0$. The monotonicity of $v(s)$ is obvious from Equation (1) that flipping a 0 bit to a 1 bit will always yield a greater value. For any $s - 2^{-N} \le s' \le s + 2^{-N}$, we specify $n_1$ and $n_0$ such that $s - 2^{-n_1} \le s' \le s + 2^{-n_0}$. By the monotonicity of $v(s)$ we have

$$v(s) - v(s') \le v(s) - v(s - 2^{-n_1})$$

$$
= (1 - p)\gamma^{n_1} \prod_{j=1}^{n_1 - 1} ((1 - p) + (2p - 1)b_j) \cdot \left(1 + \sum_{i=n_1+1}^{\infty} \gamma^{i - n_1} b_i p \prod_{j=n_1+1}^{i-1} ((1 - p) + (2p - 1)b_j)\right)
$$

$$
- (1 - p)\gamma^{n_1} \prod_{j=1}^{n_1 - 1} ((1 - p) + (2p - 1)b_j) \sum_{i=n_1+1}^{\infty} \gamma^{i - n_1} b_i (1 - p) \prod_{j=n_1+1}^{i-1} ((1 - p) + (2p - 1)b_j)
$$

$$
= (1 - p)\gamma^{n_1} \prod_{j=1}^{n_1 - 1} ((1 - p) + (2p - 1)b_j) \cdot \Bigg(1
$$

$$
+ \sum_{i=n_1+1}^{\infty} \gamma^{i - n_1} b_i (2p - 1) \prod_{j=n_1+1}^{i-1} ((1 - p) + (2p - 1)b_j)\Bigg)
$$

$$
\le (1 - p)\gamma^{n_1} p^{n_1 - 1} \cdot \left(1 + \sum_{i=n_1+1}^{\infty} \gamma^{i - n_1} (2p - 1) p^{n_1 - i - 1}\right)
$$

$$\leq 2(1-p)\gamma^N p^{N-1}.$$

And similarly,

$$v(s) - v(s') \geq v(s) - v(s + 2^{-n_0})$$

$$\geq -(1-p)\gamma^{n_0} p^{n_0 - 1} \cdot (1 + \sum_{i=n_0+1}^{\infty} \gamma^{i-n_0}(2p-1)p^{n_0 - i - 1})$$

$$\geq -2(1-p)\gamma^N p^{N-1}.$$

Hence, $|v(s) - v(s')|$ is bounded by $2(1-p)\gamma^N p^{N-1}$ for $s - 2^{-N} \leq s' \leq s + 2^{-N}$. As $2(1-p)\gamma^N p^{N-1}$ converges to zero when $N$ approaches infinity, $v(s)$ is continuous as desired.

We then show the continuity of $v'(s) = \max_{a \in \mathcal{A}(s)} Q_v(s, a)$. We first argue that $v'(s)$ is monotonically increasing. In fact, for $s' \geq s$ and $0 < a \leq \min\{s, 1 - s\}$, either $0 < a \leq \min\{s', 1 - s'\}$ or $0 < a + s - s' \leq \min\{s', 1 - s'\}$ must be satisfied. Therefore $a \in \mathcal{A}(s)$ indicates at least one of $a \in \mathcal{A}(s')$ and $a + s - s' \in \mathcal{A}(s')$.

Let $a'$ be $a$ or $a + s - s'$ whoever is in $\mathcal{A}(s')$, we have both $s' + a' \geq s + a$ and $s' - a' \geq s - a$. Specify $a$ such that $v'(s) = Q_v(s, a)$, we have

$$v'(s') \geq Q_v(s', a') \geq v'(s).$$

The monotonicity follows.

Let $s = b_1 b_2 \ldots b_\ell \ldots_{(2)}$. Similarly, for any $N$, specify $n_1 \geq N$ such that $b_{n_1} = 1$ and $n_0 \geq N + 2$ such that $b_{n_0} = 0$. Also let $s_0 = b_1 b_2 \ldots b_{N(2)}$. Then for the neighbourhood set $s_0 - 2^{-(N+1)} \leq s' \leq s_0 + 2^{-(N+1)}$, $v'(s) = v(s)$ for both the ends of the interval $s_0 - 2^{-(N+1)}, s_0 + 2^{-(N+1)} \in G_{N+1}$. $|v'(s) - v'(s')|$ is then bounded by $|v(s_0 - 2^{-(N+1)}) - v(s_0 + 2^{-(N+1)})|$. According to Claim 5, this value converges to zero when $N$ approaches infinity. The continuity of $v'(s)$ follows. $\qquad\square$

The continuity of $v(s)$ extends to the dyadic rationals $\bigcup_{\ell \geq 1} G_\ell$ when $\gamma = 1$, which means that $v(s)$ is *continuous everywhere* on $[0, 1]$ under $\gamma = 1$. It worth note that similar to the Cantor function, $v(s)$ is *not absolutely continuous*. In fact, $v(s)$ shares more common properties with the Cantor function, as they both have *a derivative of zero almost everywhere*, both their value go from 0 to 1, and their range is every value in between of 0 and 1.

The continuity of $v'(s) = \max_{a \in \mathcal{A}(s)} Q_v(s, a)$ indicates that the optimal action is uniquely $\min\{s, 1 - s\}$ on $s \notin G_\ell$. This optimal action agrees with the optimal action we specified on $s \in G_\ell$ in Lemma 8, which makes $\pi(s) = \min\{s, 1 - s\}$ an optimal policy for every state (condition on that $v(s)$ is the optimal value function, which will be proved later).

**Corollary 10.** *If $s \notin G_\ell$ for any $\ell \geq 1$,*

$$\operatorname*{argmax}_{a \in \mathcal{A}(s)} Q_v(s, a) = \{\min\{s, 1 - s\}\}.$$

**Lemma 11.** *$v(s)$ is the unique solution of the system (ABX).*

*Proof.* Let $v'(s) = \max_{a \in \mathcal{A}(s)} Q_v(s, a)$. As per Lemma 8 we have $v(s) = v'(s)$ on the dyadic rationals $\bigcup_{\ell \geq 1} G_\ell$. Since $\bigcup_{\ell \geq 1} G_\ell$ is dense and compact on $(0, 1)$, $v(s) = v'(s)$ holds whenever both $v(s)$ and $v'(s)$ are continuous at $s$. By Lemma 9 $v(s)$ and $v'(s)$ are continuous for any $s$ if there does not exist an $\ell \geq 1$ such that $s \in G_\ell$, which then indicates $v(s) = v'(s)$ on the complement of $\bigcup_{\ell \geq 1} G_\ell$. Therefore $v(s) = v'(s)$ is satisfied on $(0, 1)$, which verifies the Bellman property (AB). The boundary conditions (X) holds obviously. Finally as per Lemma 2, $v(s)$ is the unique solution to the system of Bellman equation and the boundary conditions. $\qquad\square$

**Theorem 12.** *Let $0 \leq \gamma \leq 1$ and $p > 0.5$. Under the continuous setting of the Gambler's problem, the optimal state-value function is $v(1) = 1$ and $v(s)$ defined in Equation (1) for $0 \leq s < 1$.*

*Proof.* As the optimal state-value function must satisfy the system (ABX) and $v(s)$ is the unique solution to the system, $v(s)$ is the optimal state-value function. $\qquad\square$

**Corollary 13.** *The policy $\pi(s) = \min\{s, 1 - s\}$ is (Blackwell) optimal.*

It is worth noting that when $\gamma = 1$ and $s \in G_\ell \setminus G_{\ell-1}$ for some $\ell$, then $\pi'(s) = 2^{-\ell}$ is also an optimal policy at $s$.

Theorem 12 and the proof of Lemma 8 and also induce the following statement that the optimal value function $v(s)$ is fractal and self-similar. The derivation of the corollary is in the introduction.

**Corollary 14.** *The curve of the value function $v(s)$ on the interval $[k2^{-\ell}, (k+1)2^{-\ell}]$ is similar (in geometry) to the curve of $v(s)$ itself on $[0,1]$, for any integer $\ell \geq 1$ and $0 \leq k \leq 2^\ell - 1$.*

Some other notable facts about $v(s)$ are as below:

**Fact 15.** *The expectation*
$$\int_0^1 v(s)ds = (1 - p)\gamma = v(\frac{1}{2}).$$

**Fact 16.** *The derivative*
$$\lim_{\Delta s \to 0^+} \frac{v(s + \Delta s)}{\Delta s} = 0, \quad \lim_{\Delta s \to 0^-} \frac{v(s + \Delta s)}{\Delta s} = \begin{cases} +\infty, & \text{if } s = 0 \text{ or } s \in \bigcup_{\ell \geq 1} G_\ell, \\ 0, & \text{otherwise.} \end{cases} \tag{8}$$

**Fact 17.** *The length of the arc $y = v(s)$, $0 \leq s \leq 1$ equals 2.*

In fact, any singular function (zero derivative a.e.) has an arc length of 2 (Pelling, 1977) if it go from $(0,0)$ to $(1,1)$ monotonically. This can be intuitively understood as that the curve either goes horizontal, when the derivative is zero, or vertical, when the derivative is infinity. Therefore the arc length is the Manhattan distance between $(0,0)$ and $(1,1)$, which equals 2.

**Fact 18.**
$$\operatorname*{argmin}_{0 \leq s \leq 1} v(s) - s = \{\frac{2}{3}\}.$$

It is natural that by the fractal characterization of $v(s)$ the approximation must be inexact. The following two propositions give quantitative lower bounds on such approximation errors.

**Proposition 19.** *When $N \in \mathbb{N}^+$, $N \geq 4$ is a power of 2, let $\bar{v}_1(s)$ be piecewise constant on any of the intervals $s \in (k/N, (k+1)/N)$, $k = 0, \ldots, N - 1$, then*
$$\int_s |v(s) - \bar{v}_1(s)| \, ds \geq \frac{1}{N} \frac{(2 - \gamma)(1 - p)\gamma}{1 - p\gamma} + o(\frac{1}{N}).$$

*Proof.* When $N$ is a power of 2, for $k \in \{0, \ldots, N - 1\}$, the curve of $v(s)$ on each of the intervals $(k/N, (k+1)/N)$ is self-similar to $v(s)$ itself on $(0,1)$. We consider this segment of the curve. By Equation (2),
$$v(s) = v(\bar{s}) + \gamma^\ell \prod_{j=1}^\ell ((1 - p) + (2p - 1)b_j) \cdot v(2^\ell(s - \bar{s})),$$
where $\ell = \log_2 N$, $\bar{s} = k/N$ and $s = 0.b_1 b_2 \ldots b_\ell \ldots_{(2)} \in (\bar{s}, \bar{s} + \frac{1}{N})$.

Let $\Delta s = 1/N$, $\Delta y = v((k+1)/N) - v(k/N)$, we have for $0 < s < \Delta s$
$$v(\bar{s} + s) = v(\bar{s}) + v(s/\Delta s)\Delta y.$$

As $v(s)$ is monotonically increasing on every interval $(k/N, (k+1)/N)$, the minimum over $\bar{y}$
$$\int_{s=\bar{s}}^{\bar{s}+\Delta s} |v(s) - \bar{y}| \, ds$$
is obtained when $\bar{y}_{\bar{s}} = v(\bar{s} + \frac{1}{2}\Delta s)$ (intuitively, the median of $v(s)$ on the interval). This results in an approximation error of
$$\min_{\bar{y}} \int_{s=\bar{s}}^{\bar{s}+\Delta s} |v(s) - \bar{y}| \, ds$$

$$= \int_{s=\bar{s}}^{\bar{s}+\frac{1}{2}\Delta s} v(\bar{s} + \frac{1}{2}\Delta s) - v(s) \, ds + \int_{s=\bar{s}+\frac{1}{2}\Delta s}^{\bar{s}+\Delta s} v(s) - v(\bar{s} + \frac{1}{2}\Delta s) \, ds$$

$$= -\int_{s=\bar{s}}^{\bar{s}+\frac{1}{2}\Delta s} v(s) \, ds + \int_{s=\bar{s}+\frac{1}{2}\Delta s}^{\bar{s}+\Delta s} v(s) \, ds$$

$$= -\frac{1}{2}\Delta s \int_{s=0}^{1} v(\bar{s}) + (v(\bar{s} + \frac{1}{2}\Delta s) - v(\bar{s}))v(s) \, ds$$

$$+ \frac{1}{2}\Delta s \int_{s=0}^{1} v(\bar{s} + \frac{1}{2}\Delta s) + (v(\bar{s} + \Delta s) - v(\bar{s} + \frac{1}{2}\Delta s))v(s) \, ds$$

$$= \frac{1}{2}\Delta s((1 - (1-p)\gamma)(v(\bar{s} + \frac{1}{2}\Delta s) - v(\bar{s})) + (1-p)\gamma(v(\bar{s} + \Delta s) - v(\bar{s} + \frac{1}{2}\Delta s))).$$

This error is then summed over $\bar{s} = 0, 1/N, \dots, (N-1)/N$ such that

$$\sum_{\bar{s}=0/N}^{(N-1)/N} \int_{s=\bar{s}}^{\bar{s}+\Delta s} |v(s) - \bar{y}_{\bar{s}}| \, ds \geq \frac{1}{2N}((1 - (1-p)\gamma)v(\frac{N-\frac{1}{2}}{N}) + (1-p)\gamma(1 - v(\frac{1}{2N})))$$

$$= \frac{1}{2N}((1 - (1-p)\gamma)\frac{(1-p)\gamma}{1 - p\gamma}(1 - (p\gamma)^{\log_2 N + 1})$$

$$+ (1-p)\gamma(1 - ((1-p)\gamma)^{\log_2 N + 1}))$$

$$= N^{-1}\frac{(2-\gamma)(1-p)\gamma}{1 - p\gamma} - N^{-1+\log_2 p\gamma}\frac{(1 - (1-p)\gamma)p(1-p)\gamma^2}{2(1 - p\gamma)}$$

$$- N^{-1+\log_2(1-p)\gamma}\frac{(1-p)^2\gamma^2}{2}$$

$$= \frac{1}{N}\frac{(2-\gamma)(1-p)\gamma}{1 - p\gamma} + o(\frac{1}{N}). \qquad \square$$

An error bound in $\mathcal{O}(1/N)$ can be generated to any integer $N$, as we can relax $N$ to $2^{\lfloor \log_2 N \rfloor - 1}$ so that at least one self-similar segment of size $1/N$ is included in each interval.

For Lipschitz continuous functions like neural networks, the following proposition shows an approximation error lower bound in $\mathcal{O}(1/L)$, where $L$ is the Lipschitz constant.

**Proposition 20.** *Let $L \geq (1-p)\gamma(1-\gamma)/(1-p\gamma)$. If $\bar{v}_2(s)$ is Lipschitz continuous on $s \in (0,1)$ where $L$ is the Lipschitz constant, then*

$$\int_s |v(s) - \bar{v}_2(s)| \, ds \geq \frac{1}{L}\frac{(1-p)^2\gamma^2(1-\gamma)^2}{4(1-p\gamma)}.$$

*Proof.* We consider $v(\frac{1}{2}) = (1-p)\gamma$ and

$$\lim_{s \to \frac{1}{2}^-} v(s) = \frac{(1-p)^2\gamma^2}{1 - p\gamma}.$$

When $0 < \gamma < 1$, we have

$$v(\frac{1}{2}) - \lim_{s \to \frac{1}{2}^-} v(s) = \frac{(1-p)\gamma(1-\gamma)}{1 - p\gamma} > 0.$$

Denote $h = (1-p)\gamma(1-\gamma)/(1-p\gamma)$. By the monotonicity of $v(s)$, using $\bar{v}_2(s)$ to approximate $v(s)$ has an error at least $\int_s |\xi(s) - \bar{v}_2(s)| \, ds$, where $\xi(s)$ denotes the step function on $[0,1]$,

$$\xi(s) = \begin{cases} 0 & 0 \leq s < \frac{1}{2}, \\ h & \frac{1}{2} \leq s \leq 1. \end{cases}$$

In this case, the optimal $\bar{v}_2(s)$ is

$$\bar{v}_2(s) = \begin{cases} 0 & 0 \leq s < \frac{1}{2} - \frac{h}{2L}, \\ \frac{h}{2} + L(s - \frac{1}{2}) & \frac{1}{2} - \frac{h}{2L} \leq s \leq \frac{1}{2} + \frac{h}{2L}, \\ h & \frac{1}{2} + \frac{h}{2L} < s \leq 1. \end{cases}$$

Hence, we have

$$\int_s |v(s) - \bar{v}_2(s)| \, ds \geq \int_s |\xi(s) - \bar{v}_2(s)| \, ds \geq \frac{h^2}{4L},$$

as desired. □

## A.2 ANALYSIS OF THE BELLMAN EQUATION

We have proved that $v(s)$ is the optimal value function in Theorem 12, by showing the existence and uniqueness of the solution of the system (ABX). However, the condition (X) is derived from the context of the Gambler's problem. It is rigorous enough to find the optimal value function, but we are also interested in solutions purely derived from the MDP setting. Also, algorithmic approaches such as Q-learning (Watkins & Dayan, 1992; Baird, 1995; Mnih et al., 2015) optimize the MDP by solving the Bellman equation, without eliciting the context of the problem. Studying such systems will help to understand the behavior of these algorithms. In this section, we inspect the system of Bellman equation (AB) of the Gambler's problem. We aim to solve the general case (ABY) where $p > 0.5$ and the corner case (ABZ) where $p = 0.5$.

When $p > 0.5$, the value function $v(s)$ is obviously still a solution of the system (ABY) without condition (X). The natural question is if there exist any other solutions. The answer is two-fold: When $\gamma < 1$, $f(s) = v(s)$ is unique; when $\gamma = 1$, the solution is either $v(s)$ or a constant function at least 1. This indicates that algorithms like Q-learning have constant functions as their set of converging points, apart from $v(s)$. As $v(s)$ is harder to approximate due to the non-smoothness, a constant function in fact induces a smaller approximation error and thus has a better optimality for Q-learning with function approximation.

It is immediate to generate this result to general MDPs, as function of a large constant solves MDPs with episodic rewards. This indicates that Q-learning may have more than one converging points and may diverge from the optimal value function under $\gamma = 1$. This leads to the need of $\gamma$, which is artificially introduced and biases the learning objective. More generally, the Bellman equation may have a continuum of finite solutions in an infinite state space, even with $\gamma < 1$. Some studies exist on the necessary and sufficient conditions for a solution of the Bellman equation to be the value function (Kamihigashi & Le Van, 2015; Latham, 2008; Harmon & Baird III, 1996), though the majority of this topic remains open.

The discussions above are supported by a series of rigorous statements. We begin with the following proposition that when the discount factor is strictly less than 1, the solution toward the Bellman equation is the optimal value function.

**Proposition 21.** *When $\gamma < 1$, $v(s)$ is the unique solution of the system (ABY).*

*Proof.* The uniqueness has been shown in Lemma 2 for the system (ABX). When $\gamma < 1$ it corresponds to case (I), where neither the upper bound $f(s) \leq 1$ nor the continuity at $s = 0$ in condition (X) is used. Therefore Lemma 2 holds for (ABY) under $\gamma < 1$, so follows Lemma 11 the uniqueness as desired. □

This uniqueness no longer holds under $\gamma = 1$.

**Theorem 22.** *Let $\gamma = 1$, $p > 0.5$, and $f(\cdot)$ be a real function on $[0, 1]$. $f(s)$ satisfies the Bellman equation (ABY) if and only if either*

- *$f(s)$ is $v(s)$ defined in Theorem 12, or*

- *$f(0) = 0$, $f(1) = 1$, and $f(s) = C$ for all $0 < s < 1$, for some constant $C \geq 1$.*

*Proof.* It is obvious that both $f(s)$ defined above are the solutions of the system. It amounts to show that they are the only solutions.

Without the bound condition (X), the function $f(s)$ is not necessarily continuous on $s = 0$ and $s = 1$ and is not necessarily monotonic on $s = 1$. Therefore the same arguments in the proof of Lemma 2 will not hold. However, the arguments can be extended to (Y) by considering the limit of $f(s)$ when $s$ approaches 0 and 1.

By Lemma 4 the function is continuous on the open interval $(0, 1)$. Let

$$C_0 = \lim_{s \to 0^+} f(s), \quad C_1 = \lim_{s \to 1^-} f(s).$$

Then by Lemma 3, $0 \le C_0 \le f(s) \le C_1$ for $s \in (0, 1)$. Here we eliminate the possibility of $C_0 = +\infty$ and $C_1 = +\infty$. This is because if there is a sequence of $s_t \to 0$ such that $f(s_t) > t$, then we have $f(\frac{1}{2}) \ge p\, f(s_t) + (1-p)\, f(1-s_t) \ge (1-p)t$ for any $t$. Then $f(\frac{1}{2})$ does not exist. Similar arguments show that $C_1$ cannot be $+\infty$.

Now specify a sequence $a_t \to \frac{1}{2}$, $a_t < \frac{1}{2}$, such that $C_0 \le f(\frac{1}{2} - a_t) \le C_0 + \frac{1}{t}$ and $C_1 - \frac{1}{t} \le f(\frac{1}{2} + a_t) \le C_1$. Then we have

$$f(\frac{1}{2}) \ge p\, f(\frac{1}{2} - a_t) + (1-p)\, f(\frac{1}{2} + a_t)$$

$$\ge p\, C_0 + (1-p)\, C_1 - \frac{1}{t}.$$

As $t$ is arbitrary we have $f(\frac{1}{2}) \ge pC_0 + (1-p)C_1$. By induction on $\ell$ it holds on $s \in \bigcup_{\ell \ge 1} G_\ell$ that

$$f(s) \ge C_0 + (C_1 - C_0)v(s).$$

By Lemma 4 the continuity of $f(s)$ and $v(s)$ under $\gamma = 1$, this lower bound extends beyond the dyadic rationals to the entire interval $(0, 1)$. Define $\tilde{f}(s) = C_0 + (C_1 - C_0)v(s)$ for $s \in (0, 1)$, $\tilde{f}(0) = C_0, \tilde{f}(1) = C_1$. It is immediate to verify that for any $C_1 > C_0 \ge 0$, $\tilde{f}(s)$ solves the system (AY) (without (B) the boundary conditions).

If $C_1 - C_0 \ne 0$, by Lemma 2 Case (II) $\tilde{f}(s)$ on $(0, 1)$ is the unique solution of the system (AY), given monotonicity, continuity, and the lower bound $\tilde{f}(s)$. With the boundary conditions (B), we have $0 = \tilde{f}(0) = C_0$ and $1 = \tilde{f}(1) = C_1$, therefore $f(s) = v(s)$. This case $C_1 - C_0 \ne 0$ induces the first possible solution.

It amounts to determine $f(s)$ when $C_1 - C_0 = 0$, that is, when $0 \le C_0 = f(s) = C_1$ for $s \in (0, 1)$ (by Lemma 3). It suffices to prove that $C_0 < 1$ is not possible. In fact, if $C_0 < 1$ then $f(\frac{3}{4}) < p\, f(\frac{1}{2}) + (1-p)f(1)$, which contradicts (A). Then, $f(s) = C_0$ for some $C_0 \ge 1$ is the only set of solutions when $C_1 - C_0 = 0$, as desired. $\qquad \square$

The fact that a large constant function can also be a solution toward the Bellman equation can be extended to a wide range of MDPs. The below proposition lists one of the sufficient conditions but even without this condition it is likely to hold in practice.

**Proposition 23.** *For an arbitrary MDP with episodic rewards where every state has an action to transit to a non-terminal state almost surely, $f(s) = C$ for all non-terminal states $s$ is a solution of the Bellman equation system for any $C$ greater or equal to the maximum one-step reward.*

*Proof.* The statement is immediate by verifying the Bellman equation. $\qquad \square$

The rest of the section discusses the Gambler's problem under $p = 0.5$, where the gambler does not lose capital by betting in expectation. In this case, the optimal value function is still $v(s)$ by similar arguments of Theorem 12. It is worth noting that when $\gamma = 1$, Theorem 12 indicates $v(s) = s$. This agrees with the intuition that the gambler does not lose their capital by placing bets in expectation, therefore the optimal value function should be linear to $s$. Proposition 21 also holds that $v(s)$ is the unique solution of the Bellman equation, given $\gamma < 1$. The remaining problem is to find the solution of the Bellman equation under $\gamma = 1$ and $p = 0.5$. This corresponds to the system (ABZ).

When $p = 0.5$, condition (A) implies midpoint concavity such that for all $a \in \mathcal{A}(s)$,

$$f(s) \ge \frac{1}{2}f(s - a) + \frac{1}{2}f(s + a), \tag{9}$$

where the equality must hold for some $a$. As Lemma 3 no longer holds, a solution $f(s)$ may have negative values for some $s$. Though, if it does not have a negative value, it must be concave, and thus linear by condition (A) (will be proved later in Theorem 27). It suffices to satisfy $f(s) \ge s$ for any

$s$. Therefore the non-negative solution is $f(0) = 0$, $f(1) = 1$, and $f(s) = C's + B'$ on $0 < s < 1$ for some constants $C' + B' \geq 1$.

If $f(s)$ does have a negative value at some $s$, then the midpoint concavity (9) does not imply concavity. In this case, by recursively applying (9) we can show that the set $\{(s, f(s)) \mid s \in (0, 1)\}$ is dense and compact on $(0, 1) \times \mathbb{R}$. Then the function becomes pathological, if it exists. Despite this, the following lemma shows that $f(s)$ need to be positive on the rationals $\mathbb{Q}$.

**Lemma 24.** *Let $f(s)$ satisfies (ABZ). If there exists $0 \leq s^- < s^+ \leq 1$ and a constant $C$ such that $f(s^-), f(s^+) \geq C$, then $f(s) \geq C$ for all $s \in \{s^- + w(s^+ - s^-) \mid w \in \mathbb{Q}, 0 \leq w \leq 1\}$.*

*Proof.* The statement is immediate for $w \in \{0, 1\}$. For $0 < w < 1$ we prove the lemma by contradiction. Let $f(s^- + w(s^+ - s^-)) < C$ for some $w \in \mathbb{Q}$ while $0 < w < 1$. We define $s_0 = s^- + w(s^+ - s^-)$ and $s_{t+1} = 2s_t - s^-$ for $s_t < \frac{1}{2}(s^- + s^+)$ and $s_{t+1} = 2s_t - s^+$ for $s_t > \frac{1}{2}(s^- + s^+)$, respectively. $s_{t+1}$ will be undefined if $s_t = \frac{1}{2}(s^- + s^+)$. Since $w \in \mathbb{Q}$, let $w = m/n$ where $m$ and $n$ are integers and the greatest common divisor $\gcd(m, n) = 1$. Then $(s_t - s^-)/(s^+ - s^-) = m_t/n$, where $m_t = 2^t m \bmod n$. As $\mathbb{Z}_n$ is finite, $\{s_t\}_{t \geq 0}$ can only take finite many values. Thus either the sequence $\{s_t\}$ is periodic, or it terminates at some $s_t = \frac{1}{2}(s^- + s^+)$.

Then we show that $f(s_t)$ is strictly decreasing by induction. Assume that $f(s_0) > \cdots > f(s_t)$. When $s_t < \frac{1}{2}(s^- + s^+)$, by (9) we have $f(s_t) \geq \frac{1}{2}f(s^-) + \frac{1}{2}f(s_{t+1})$, which indicates that $f(s_{t+1}) - f(s_t) \leq f(s_t) - f(s^-) < f(s_0) - f(s^-) < 0$. When $s_t > \frac{1}{2}(s^- + s^+)$, by (9) we have $f(s_t) \geq \frac{1}{2}f(s_{t+1}) + \frac{1}{2}f(s^+)$, which indicates $f(s_{t+1}) - f(s_t) \leq f(s_t) - f(s^+) < f(s_0) - f(s^+) < 0$. The base case $f(s_1) < f(s_0)$ holds as at least one of $f(s_0) \geq \frac{1}{2}f(s^-) + \frac{1}{2}f(s_1)$ and $f(s_0) \geq \frac{1}{2}f(s_1) + \frac{1}{2}f(s^+)$ must be true. Thus we conclude that $f(s_t)$ is strictly decreasing.

If the sequence terminates at some $s_t = \frac{1}{2}(s^- + s^+)$, then $f(s_t) < f(s_1) < C$, which contradicts $f(s_t) = f(\frac{1}{2}(s^- + s^+)) \geq \frac{1}{2}f(s^-) + \frac{1}{2}f(s^+) \geq C$. Otherwise $s_t$ is periodic and indefinite. Denote the period as $T$ we have $f(s_{t+T}) < f(s_t)$, which indicates $f(s_t) < f(s_t)$ as a contradiction. $\square$

Lemma 24 agrees with the statement that the midpoint concavity indicates rational concavity. The below statements give some insights into the irrational points.

**Lemma 25.** *Let $f(s)$ satisfies (ABZ). If there exists an $\bar{s} \in \mathbb{R} \setminus \mathbb{Q}$ such that $f(\bar{s}) \geq 0$, then $f(s) \geq 0$ for all $s \in \{w\bar{s} + u \mid w, u \in \mathbb{Q}, 0 \leq w, u, \leq 1, w + u \leq 1\}$.*

*Proof.* Specify $s^- = \bar{s}$ and $s^+ = 1$ in Lemma 24, we have $f(\bar{s} + \frac{u}{w+u}(1 - \bar{s})) \geq 0$ whenever $0 \leq \frac{u}{w+u} \leq 1$ and $\frac{u}{w+u} \in \mathbb{Q}$. This is satisfied when $0 \leq w, u \leq 1$, $w + u > 0$, $w, u \in \mathbb{Q}$. Specify $s^- = 0$ and $s^+ = \bar{s} + \frac{u}{w+u}(1 - \bar{s})$ in Lemma 24, we have $f(w\bar{s} + u) = f((w + u)(\bar{s} + \frac{u}{w+u}(1 - \bar{s}))) \geq 0$ whenever $w + u \leq 1$. Thus $f(w\bar{s} + u) \geq 0$ for $0 < w, u < 1$, $w, u \in \mathbb{Q}$, and $0 < w + u \leq 1$. Since the case $w = u = 0$ is immediate, the statement follows with $s \in \{w\bar{s} + u \mid w, u \in \mathbb{Q}, 0 \leq w, u, \leq 1, w + u \leq 1\}$. $\square$

**Corollary 26.** *Let $f(s)$ satisfies (ABZ). If there exists an $\bar{s} \in \mathbb{R} \setminus \mathbb{Q}$ such that $f(\bar{s}) < 0$, then $f(w\bar{s})$ is monotonically decreasing with respect to $w$ for $w \in \mathbb{Q}$, $1 \leq w < 1/\bar{s}$.*

Lemma 25 and Corollary 26 indicate that when there exists a negative or positive value, infinitely many other points (that are not necessarily in its neighbor) must have negative or positive values as well. It is intuitive that the solution with a negative value, if it exists, must be complicated and pathological. In fact, Sierpiński has shown that a midpoint concave but non-concave function is not Lebesgue measurable and is non-constructive (Sierpiński, 1920a;b), so does $f(s)$ if it solves (ABZ) while having a negative value.

Such an $f(s)$ exists if and only if we assume Axiom of Choice (Jech, 2008; Sierpiński, 1920a;b). We consider the vector space by field extension $\mathbb{R}/\mathbb{Q}$. With the axiom specify a basis $\mathbb{B} = \{1\} \cup \{g_i\}_{i \in \mathcal{I}}$, known as a Hamel basis. With this basis $\mathbb{B}$ every real number can be written uniquely as a linear combination of the elements in $\mathbb{B}$ with rational coefficients. Therefore, denote a real number $s$ uniquely as a vector $(w, w_i)_{i \in \mathcal{I}}$, $w, w_i \in \mathbb{Q}$, such that $s = w + \sum_{i \in \mathcal{I}} w_i g_i$. We correspond each $f(s) = f'(w, \{w_i\}_{i \in \mathcal{I}})$ uniquely to $f'$ defined on $\mathbb{Q}^{|\mathbb{B}|}$ and use the two spaces interchangeably.

The solution $f(s)$ to (9) is any concave function $f'$ on the vector space $\mathbb{R}/\mathbb{Q}$. Based on this solution we extend the system (9) to (ABZ). To this end, $f(s)$ need to attain the equality in (9) at every $s$, which holds if and only if for every $s = w + \sum_{i \in \mathcal{I}} w_i g_i$ there exists $s^1 = w^1 + \sum_{i \in \mathcal{I}} w_i^1 g_i$ and $s^2 = w^2 + \sum_{i \in \mathcal{I}} w_i^2 g_i$ such that $f'(\lambda w^1 + (1-\lambda)w^2, \{\lambda w_i^1 + (1-\lambda)w_i^2\}_{i \in \mathcal{I}})$ is $\lambda f(s^1) + (1-\lambda)f(s^2)$ for any $0 < \lambda < 1$ (intuitively, local linearity of $f'$ on at least one direction everywhere). By specifying a $\mathbb{B}$, the condition can be met if $f(s) = f'(w, \{w_i\}_{i \in \mathcal{I}}) = \alpha(w_j) + \beta(\overline{w}_j)$ for some $w_j \in \{w\} \cup \{w_i \mid i \in \mathcal{I}\}$, where $\alpha$ is a linear function, $\beta$ is a concave function, and $\overline{w}_j$ denotes $\{w\} \cup \{w_i \mid i \in \mathcal{I}\} \setminus \{w_j\}$. When $w_j$ is $w$, $f(s)$ is in fact $w + \beta(\{w_i\}_{i \in \mathcal{I}})$ for some concave function $\beta$. This is equivalent to specifying a function $\omega(s) : \mathbb{R} \to \mathbb{Q}$ such that $\omega$ maps reals to rationals additively $\omega(s_1 + s_2) = \omega(s_1) + \omega(s_2)$ and $\omega$ is not constantly zero, and then write $f(s)$ as $\omega(s) + \beta_1(s - \omega(s))$ for some concave real function $\beta_1$. Otherwise if $w_j$ is not $w$, $f(s)$ is in the aforementioned form with the boundary conditions (B).

While we have shown that under Axiom of Choice $f(s) = \alpha(w_j) + \beta(\overline{w}_j)$ is a set of solutions that can be described by the infinite-dimensional vector space $\mathbb{R}/\mathbb{Q}$ and its basis, we do not know if they are the only solutions. Nevertheless, combining our analysis and the literature we conclude the following statement about the system (ABZ).

**Theorem 27.** *Let $\gamma = 1$ and $p = 0.5$. A real function $f(s)$ satisfies (ABZ) if and only if either*

- *$f(s) = C's + B'$ on $s \in (0, 1)$, for some constants $C' + B' \geq 1$, or*

- *$f(s)$ is some non-constructive, not Lebesgue measurable function under Axiom of Choice.*

*Proof.* The first bullet correspond to the function $f(s)$ that is non-negative on $[0, 1]$. By the midpoint concavity (9) and the fact that $f(s)$ is non-negative, $f(s)$ is concave on $[0, 1]$ (Sierpiński, 1920a;b). We specify $s_0 = \frac{1}{2}$. By (A) we have

$$f(s_0) = \frac{1}{2}f(s_0 - a_0) + \frac{1}{2}f(s_0 + a_0)$$

for some $a_0$. Since $f(s)$ is concave, $f(s)$ must be linear on the interval $[s_0 - a_0, s_0 + a_0]$. Consider the nonempty set

$$\mathcal{A}_1 = \left\{a_0 \in \mathcal{A}(s_0) \;\middle|\; f(s_0) = \frac{1}{2}f(s_0 - a_0) + \frac{1}{2}f(s_0 + a_0)\right\}.$$

We show by contradiction that $\sup \mathcal{A}_1$ is $\frac{1}{2}$. If $a_1 = \sup \mathcal{A}_1 < \frac{1}{2}$, then by the continuity $a_1 \in \mathcal{A}_1$, where the continuity is implied by the convexity. This indicates that $f(s) = f(s_0) + \frac{f(s_0 + a_1) - f(s_0 - a_1)}{2a_1}(s - s_0)$ when $s_0 - a_1 \leq s \leq s_0 + a_1$. But as $a_1$ is the maximum element of $\mathcal{A}_1$, on at least one of the intervals $[0, s_0 - a_1)$ and $(s_0 + a_1]$ we have by the convexity $f(s) < f(s_0) + \frac{f(s_0 + a_1) - f(s_0 - a_1)}{2a_1}(s - s_0)$. Therefore, for at least one of $s \in \{s_0 - a_1, s_0 + a_1\}$ we have $f(s) > \frac{1}{2}f(s - a) + \frac{1}{2}f(s + a)$ for all $a$, which contradicts condition (A). Hence $\sup \mathcal{A}_1$ is $\frac{1}{2}$, which implies that $f(s)$ is linear on $(0, 1)$. Writing $f(s)$ as $C's + B'$, by the boundary condition (B) we have $C' + B' \geq 1$. It is immediate to verify that $f(s) = C's + B'$ with $C' + B' \geq 1$ is sufficient to satisfy the system (ABZ), and is thus the non-negative solution of the system.

If $f(s)$ is negative at some $s$, then by (9) and (B) $f(s)$ must not be concave. By Sierpiński (1920a;b) such an $f(s)$ exists only if we assume Axiom of Choice, and is non-constructive and not Lebesgue measurable if it exists (some discussion on this function is given above the statement of this theorem). □

