# OpenReview forum: "The Gambler's Problem and Beyond"
_ICLR.cc/2020/Conference — Accept (Poster)_

### Official Review · AnonReviewer1 · 2019-10-23
**Official Blind Review #1**

**Rating:** 6

**Review:**

The paper revisits the Gambler's problem. It studies a generalized formulation with continuous state and action space and shows that the optimal value function is self-similar, fractal and non-rectifiable. That is, it cannot be described by any simple analytic formula. Based on this, it also deeply analysis the discrete case.

Overall, the paper is extremely well written. I must admit that i have really enjoyed reading it. However, the discussion of the implications falls too short. There are a number of complexity results for MDPs and POMDPS, see e.g. (Papadimitriou, Tsitsiklis 1997; Lusena et al. JAIR 2001; Lee at al NIPS 2007). Indeed, the paper does not hide this and provides references. However, for an informed outsider such as the reviewer it does argue how exactly it extends our knowledge here.

**Experience Assessment:**

I have published one or two papers in this area.

**Review Assessment: Checking Correctness Of Derivations And Theory:**

I assessed the sensibility of the derivations and theory.

**Review Assessment: Checking Correctness Of Experiments:**

I carefully checked the experiments.

**Review Assessment: Thoroughness In Paper Reading:**

I read the paper at least twice and used my best judgement in assessing the paper.

---

> ### Author Response · Authors · 2019-11-07
> **Review response**
>
> We thank the reviewer for the encouraging review and for their really enjoy reading our manuscript! Indeed, it is enjoyable to explore uncharted problems and share our discovery with the community. We believe it is beneficial to have the community to be able to read our paper and also enjoy these new findings.
>
> We have made a major revision to help potential readers in the community to understand the paper and get more value out of it. The presentation has been largely improved throughout the manuscript. The implication has been extended to a section that includes rigorous statements on value function approximation, which is one of the topics that may bring some insights to the community. Other results are also backed by our theorems that are explicitly pointed to. Another notable change is that more references are added along with the discussion, including papers from reinforcement learning algorithms, MDPs, and mathematics. Armed with the revision, the informed outsiders may have a better shape to understand our work and really get inspired by the new findings. Thus we believe we are on the right track to get the paper publishable.

---

> > ### Comment · AnonReviewer1 · 2019-11-12
> > **Extended Discussion of Implications**
> >
> > Thanks for extending the discussion of the implications. However, again for an informed outsider such as the reviewer, can you actually mention explicitly what you add on top of the existing related work? The hardness of RL is already well known. The problem of finding a good approximation (and whether this can actually be done) has also been addressed. So what does your statement "representation of such function must be inexact" add to the state-of-the-art? Similar for the other parts of the paper.

---

> > > ### Author Response · Authors · 2019-11-13
> > > **Implications are not (necessarily) immediate**
> > >
> > > Thanks for further commenting on our manuscript. The hardness of RL is indeed already well known, but it is not known by the community that it can be as complex and pathological as a Cantor function, while this is likely generated to a large family of RL problems. In fact, hardness is mostly observed empirically, such as sample complexity, approximation error, empirical convergence rate, and etc. Our paper discusses *why* the hardness happens, similar to what [1] discusses on the complexity of MDPs.
> > >
> > > As an example, "representation of such function must be inexact" is indeed observed for years, but why it is inexact? Even it is handled by algorithms and heuristics on some famous tasks, algorithms may fail on some other tasks and new algorithms can be brought out by new insights.
> > >
> > > We extend the implications and hope to bring more insights into possible future algorithms and applications. Our results are new and might be surprising: They are new things to the community and are orthogonal to what people already know. It is reasonable to believe that it will introduce ideas for future works.
> > >
> > > It is very natural to expect an algorithmic approach derived by our theorems, or even a use case, but they are unlikely to be with this specific manuscript. The derivation of the theorems itself takes 20 pages, and adding more objectives to this manuscript seems too heavy from our perspective.
> > >
> > > [1] The Complexity of Markov Decision Processes, Papadimitriou 1987

---

### Official Review · AnonReviewer3 · 2019-10-23
**Official Blind Review #3**

**Rating:** 6

**Review:**

The paper derives the optimal value function for the continuous and discrete versions of the Gambler's problem. It also explores certain properties of this value function and finds that it has many fractal-like qualities.

This paper focuses on an unconventional but intriguing topic, bringing more insight into the possible shapes of value functions. The main result, which describes the value function for the gambler's problem, seems surprisingly complex and could have some practical implications for value-learning. Unfortunately, I find the paper is difficult to follow due its organization and the presence of many typos and unclear phrases. In summary, while I believe the topic is interesting and the work seems sound, the presentation would need to be improved significantly for me to recommend acceptance.

Here are certain points to be addressed:
1) On p.2, "The optimal value function presents its self-similar, fractal and non-rectifiable form". It is not clear from Thm. 11 that these properties hold. Some further explanation would be helpful
2) I think it would be helpful to include some descriptions of certain concepts that may be less familiar to the reader in the appendix. E.g. the Cantor function
3) Fig.1 should have larger labels and dots. It is difficult to see currently.
4) p.3 "the similar fractal patterns observed empirically in other reinforcement learning tasks." Certain concrete examples would be helpful here since I was not aware that this was common.
5) p.4 The paragraph on 'self-similarity' was not clear to me. What is meant by 'chaos' or 'dimension' in this context?
6) p.5 "Otherwise an arbitrary small amount of will have a fixed probability of reaching the target 1." I am unsure what this sentence means.
7) Perhaps starting with the intuitive description of v(s) (the recursive form) first before presenting the closed form solution would be easier to follow.
8) Why is there no discussion/conclusion section at the end?

There are numerous typos scattered throughout the paper. Here are some from the first page:
- abstract: "where they mention an interesting pattern of
the optimal value function with high-frequency components and repeating nonsmooth
points but without further investigation." Awkward phrase
- abstract: "With the analysis," Unnecessary phrase
- par. 1: "is described as below" -> "is described below"
- "which would hide its attractiveness." Awkward phrase
- "as an representative" -> "as a representative"
- "family of Markov decision process" -> "family of Markov decision processes"
- "the amount of bet." -> "the amount bet."
- "a round of bet." -> "a round of betting."
- "action-state value function" -> "state-action value function"
- "n as the starting capital (n denotes the state in the discrete setting)," The phrase is a bit confusing as it suggests n denotes both the discrete state and the starting capital.


**Experience Assessment:**

I have published one or two papers in this area.

**Review Assessment: Checking Correctness Of Derivations And Theory:**

I assessed the sensibility of the derivations and theory.

**Review Assessment: Checking Correctness Of Experiments:**

N/A

**Review Assessment: Thoroughness In Paper Reading:**

I read the paper at least twice and used my best judgement in assessing the paper.

---

> ### Author Response · Authors · 2019-11-07
> **Review response**
>
> We thank the reviewer for their detailed and encouraging review. Your review is very helpful for us to revise the manuscript. A revision has been updated in the system that reflects these changes.
>
> The topic of our manuscript is indeed unconventional but very intriguing. If the presentation is the most concern on our manuscript, we believe it can be improved rather quickly with the reviewers' help to be publishable.
>
> We have addressed the raised concerns:
>
> 2), 3), 5), and the typos, are addressed as pointed out.
>
> 1) On p.2, "The optimal value function presents its self-similar, fractal and non-rectifiable form". It is not clear from Thm. 11 that these properties hold. Some further explanation would be helpful
>
> We added a note pointing to the exact description of the self-similarity, in Corollary 13 (which is a corollary of Lemma 5's proof).
>
> 4) p.3 "the similar fractal patterns observed empirically in other reinforcement learning tasks." Certain concrete examples would be helpful here since I was not aware that this was common.
>
> A concrete example we know is Mountain Car (https://gym.openai.com/envs/MountainCar-v0/), where when the optimal value function v(s) is plotted on the two-dimensional state space by a heat map, high-frequency and fractal pattern can be observed. This observation in fact was heavily discussed during a seminar the I attended by the curiosity of the audience.  Though this observation is empirical by plotting and zooming in, it is convincing enough that v(s) is likely fractal. When the dimension is greater than 2 we are not sure how this can be observed even empirically.
>
> We have temporarily commented out this statement until more examples are founded. But intuitively, when simple problems like Gambler and Mountain Car to have this level of complexity, there suppose to be a family of MDPs that shares the same level of complexity.
>
> 6) p.5 "Otherwise an arbitrary small amount of will have a fixed probability of reaching the target 1." I am unsure what this sentence means.
>
> This means if the function is nonzero when s approaches 0^+, say the limit is C, then the gambler starting with \eps capital can reach the target capital of 1 with probability at least C. Then the expected capital increased from \eps to at least C * 1 = C, which contradicts with the fact p > 0.5 so that expectation must decrease as the game goes.
>
> This is revised and explained in our revised manuscript.
>
> 7) Perhaps starting with the intuitive description of v(s) (the recursive form) first before presenting the closed form solution would be easier to follow.
>
> We have added a sentence that points to the intuitive description before introducing the Theorem 11, so that the reader can find the intuitive description whenever they want so.
>
> 8) Why is there no discussion/conclusion section at the end?
>
> We have added a conclusion/future work section. Indeed, it is important to have the section to better position our paper and point out the possible future works.

---

> > ### Comment · AnonReviewer3 · 2019-11-15
> > **Adjusted score**
> >
> > Thank you for tidying up the presentation and for the clarifications, the paper is easier to follow now.
> > I have revised my score accordingly.
> >
> > Typo in the sentence "Otherwise an arbitrary small amount of will have a fixed probability of reaching the target 1." -> "amount of capital"

---

> > > ### Author Response · Authors · 2019-11-15
> > > **Thank you**
> > >
> > > We thank the reviewer very much. It is encouraging to see that our manuscript is now easier to follow as well the increased score. We have also fixed the typo and added the additional notes to our draft.

---

> ### Author Response · Authors · 2019-11-09
> **Additional notes on 4) and 6)**
>
> We thank the reviewer again for providing their helpful comments. Below are additional notes on our response.
>
> On 4): It is convincing and illustrative from 6:35 and 33:20 of the video [1] that the optimal value function of Mountain Car is fractal and self-similar on a substantial region of the state space. As we stated before, we believe that this property extends beyond Gambler and Mountain Car.
>
> On 6): We would like to note that technically, it is *not* important to have this assumption just to solve the problem. Without this assumption, Section 4.1 (i.e. system (ABX)) will give two solutions: v(s), and f(0)=1, f(s)=1 otherwise. It is immediate to argue that the later is not the optimal value function. Then the former must be. We put this assumption to delay this "f(0)=1, f(s)=1 otherwise" solution to Section 4.2 only for a better organization of our manuscript.
>
> [1] The role of interest in prediction and control. https://youtu.be/aFXdpCDAG2g. Valliappa Chockalingam et al 2019.

---

> > ### Comment · AnonReviewer3 · 2019-11-15
> > **Thanks**
> >
> > Thanks for the value function example, I was not aware of this.

---

### Official Review · AnonReviewer2 · 2019-11-01
**Official Blind Review #2**

**Rating:** 6

**Review:**


The paper gave a very detailed analysis of the value function of the gamblers problem. In this problem, the agent decides how much to bet in each step, doubling or losing the chosen amount. There is a chance of loss of greater than 0.5 and as a result the agent would like to minimize the number of steps. While one optimal policy is extremely simple (just bet the maximum), this paper shows that the value function is deceptively complicated due to its fractal structure. In particular, the value function is continuous only when there is no discounting and does not have well behaved derivatives. The exposition is clear and well structured. The author argues such behaviors have consequences for valued function based RL such as value iterations or Q-learning, since one of the simplest RL problem has a weird value function.

The paper presented both an interesting issue and a clear analysis, and should be accepted. The major issue is that any implications for RL seems unverified. For example, I would be very surprised if a tabular method will not solve the problem given a reasonable update order. The author also did not provide any analysis on how a parameterized method will actually behave, given that is the likely implication. So it is a bit unclear if the pathology is due to taking the continuous limit or if it is present in discrete cases as well. The final part relying on the axiom of choice is too mathy for the problem motivation (since it applies to the rather boring p=0.5 case). The paper would be more valuable to the ML community if that part is replaced with some analysis/evidence of how the issue shows up when actually running one of the solution methods.


**Experience Assessment:**

I have read many papers in this area.

**Review Assessment: Checking Correctness Of Derivations And Theory:**

I assessed the sensibility of the derivations and theory.

**Review Assessment: Checking Correctness Of Experiments:**

N/A

**Review Assessment: Thoroughness In Paper Reading:**

I read the paper thoroughly.

---

> ### Author Response · Authors · 2019-11-07
> **Review response**
>
> We thank the reviewer for their positive and encouraging review. We are delighted to hear that our exposition is clear and well structured. A revision has been made to improve our manuscript
>
> We understand that implication is important to our theoretical contributions. Therefore, an expanded implication section (Section 1.2) now discusses the three implications in detail. These implications are now backed by the new Proposition 27, 28, and Fact 15 and Theorem 19. The two new propositions do provide analysis on how a parameterized method will actually behave, by giving the error lower bound in O(1/N) and poly(ln L).
>
> We want to clarify that per our Proposition 1, the pathology happens to the discrete case as well. It has the same fractal and self-similar structure although discrete MDP can be solved numerically in practice. The final part is indeed mathy, so we limit their space (Lemma 21 - 24) to be less than 1 page at the end of the analysis. In this way, they maintain the completeness of our analysis of the Bellman equation. We hope that the extended implication in our revised manuscript does provide more value to the ML community.

---

### Public Comment · ~Kevin_A._Wang1 · 2023-05-26
**Cite previous works?**

Hello from the future! I like this problem and your paper is great. I think the authors and reviewers at the time missed the opportunity to cite previous analyses on this problem. For example, the book "Dubbins, Lester E and Savage, Leonard J. Inequalities for Stochastic Processes; How to Gamble If You Must. Dover Publications (1976)."

And this writeup based on the book: https://www.maa.org/sites/default/files/pdf/joma/Volume8/Siegrist/RedBlack.pdf

These previous analyses arrive to some of the same results, and their existence may contradict the conclusion's statement: "Despite its seeming simpleness, these results are not clearly pointed out in previous studies."

I hope this helps anyone who wants to find more information in the future.

---

> ### Author Response · Authors · 2023-05-26
> **Thank you for pointing out the previous works!**
>
> I agree that the writeup concludes some of the same results. We weren't able to find them during the development of our analysis. In fact, we suspected the existence of these analyses due to how "vanilla" the problem setting is. We thank you for directing us to a previous work.
>
> We take back the words "these results are not clearly pointed out in previous studies" and use this comment to refer the readers to [1] for further information.
>
> [1] Siegrist, Kyle. "How to gamble if you must." AMC 10 (2008): 12.
>
> I haven't got a chance to read the book, but will update the comment and references here once I finish so.

---

### Decision · Program_Chairs · 2019-12-19

**Decision:**

Accept (Poster)

**Comment:**

This paper studies the optimal value function for the gambler's problem, and presents some interesting characterizations thereof. The paper is well written and should be accepted.

---

> ### Public Comment · ~Kevin_A._Wang1 · 2023-05-26
> **Cite previous works?**
>
> Hello from the future! I like this problem and your paper is great. I think the authors and reviewers at the time missed the opportunity to cite previous analyses on this problem. For example, the book "Dubbins, Lester E and Savage, Leonard J. Inequalities for Stochastic Processes; How to Gamble If You Must. Dover Publications (1976)."
>
> And this writeup based on the book: https://www.maa.org/sites/default/files/pdf/joma/Volume8/Siegrist/RedBlack.pdf
>
> I hope this helps anyone who wants to find more information in the future.